# A large fraction of neocortical myelin ensheathes axons of local inhibitory neurons

Kristina D Micheva[1]*, Dylan Wolman[2], Brett D Mensh[2], Elizabeth Pax[2], JoAnn Buchanan[1†], Stephen J Smith[1], Davi D Bock[2]*

[1]Department of Molecular and Cellular Physiology, Stanford University, Stanford, United States; [2]Janelia Research Campus, Howard Hughes Medical Institute, Ashburn, United States

**Abstract** Myelin is best known for its role in increasing the conduction velocity and metabolic efficiency of long-range excitatory axons. Accordingly, the myelin observed in neocortical gray matter is thought to mostly ensheath excitatory axons connecting to subcortical regions and distant cortical areas. Using independent analyses of light and electron microscopy data from mouse neocortex, we show that a surprisingly large fraction of cortical myelin (half the myelin in layer 2/3 and a quarter in layer 4) ensheathes axons of inhibitory neurons, specifically of parvalbumin-positive basket cells. This myelin differs significantly from that of excitatory axons in distribution and protein composition. Myelin on inhibitory axons is unlikely to meaningfully hasten the arrival of spikes at their pre-synaptic terminals, due to the patchy distribution and short path-lengths observed. Our results thus highlight the need for exploring alternative roles for myelin in neocortical circuits.

*For correspondence: kmicheva@
stanford.edu (KDM); bockd@
janelia.hhmi.org (DDB)

Present address: †Allen Institute
for Brain Science, Seattle, United
States

Competing interest: See
page 23

Reviewing editor: Inna Slutsky,
Tel Aviv University, Israel

## Introduction

Myelination is well known to be important in increasing the speed of action potential propagation in long-axon connections (*Waxman and Bennett, 1972*; *Moore et al., 1978*). The axons of the central nervous system (CNS) connecting distant brain areas are typically wrapped in myelin, as are the axons of the peripheral nervous system (PNS) that connect the CNS to all skeletal muscles and many sensory neurons. Pathologies of myelinated axons are associated with numerous neurological disorders (*Fields, 2008*) and are fundamental to several, such as multiple sclerosis (MS) (*Popescu and Lucchinetti, 2012*; *Calabrese et al., 2015*), neuromyelitis optica (*Wingerchuk et al., 2015*), and leukodystrophies (*Gordon et al., 2014*).

Because myelin increases the conduction velocity of action potentials and reduces their metabolic cost (*Nave, 2010*; but see *Harris and Attwell, 2012*), it has traditionally been viewed as 'wire insulation' for long-range neural projections. Recent findings have cast myelination as a dynamic process playing an active role in normal neural circuit function and plasticity. For example, adult-born oligodendrocytes (the myelin-producing cells of the CNS) contribute new myelin to axons in an ongoing fashion (*Young et al., 2013*), increased neuronal activity results in more myelination (*Gibson et al., 2014*), and myelin remodeling in the CNS is required for motor skill acquisition (*McKenzie et al., 2014*). Myelin also supplies ensheathed axons with lactate via the monocarboxylate transporter MCT1, possibly playing an important energetic support role in the CNS (*Rinholm et al., 2011*; *Lee et al., 2012*; *Fünfschilling et al., 2012*; *Morrison et al., 2013*).

The emerging diversity of roles for myelin calls for a more complete understanding of its distribution among neuronal cell types and along individual axonal arbors, particularly in the gray matter of

**eLife digest** The brain is far away from the muscles that it controls. In humans, for example, the brain must be able to trigger the contraction of muscles that are more than a meter away. This task falls to specialized motor neurons that stretch from the brain to the spinal cord, and from the spinal cord to the muscles.

Neurons transmit information (in the form of electrical nerve impulses) along their length through cable-like structures called axons. The axons of the motor neurons are so long that, if they were 'naked', it would take at least a second for nerve impulses to travel their entire length. Such a long delay between thoughts and actions would make rapid movement impossible.

Nerve impulses are able to travel from the brain to the muscles much more quickly, because they are wrapped with a substance called myelin that acts like electrical insulation. Myelin helps the nerve impulses travel up to 100 times faster down the axon. Not surprisingly, diseases that damage myelin, such as multiple sclerosis, severely disrupt movement and sensation.

Not all of the brain's myelin is found around the long axons of motor neurons. The outer layer of the brain, known as the cerebral cortex, also contains myelin. However, most neurons within the cerebral cortex are only a few millimeters long. So what exactly is myelin doing there?

Micheva et al. have now used electron microscopy and light microscopy to study the neurons in the cortex of the mouse brain. This revealed that up to half of the myelin in some layers of the cortex surrounds the axons of inhibitory neurons (which reduce the activity of the neurons they signal to). Moreover, one particular subtype of inhibitory neuron accounts for most of the myelinated inhibitory axons, namely inhibitory neurons that contain a protein called parvalbumin.

Exactly why parvalbumin-containing neurons are myelinated remains a mystery. Myelin covers only short segments of the axons of these neurons, so it would speed up the transmission of signals by less than a millisecond – probably not enough to make a meaningful difference. Parvalbumin-containing neurons often signal frequently, and thus require large amounts of energy. One possibility therefore is that myelin helps to meet these energy requirements or to reduce energy consumption. Further research will be needed to test this and other ideas.

the CNS. Excitatory pyramidal neurons in cortex are known to be a main source of long-range myelinated axons connecting the two cortical hemispheres or targeting subcortical structures. Foundational EM-based studies of individual Golgi- or dye-filled neurons (*Kisvárday et al., 1986*; *Martin and Whitteridge, 1984*; *Peters and Proskauer, 1980*; *Tamás et al., 1997*) have revealed that the local axon collaterals of both excitatory and inhibitory neurons may also be myelinated. However, population surveys of the distribution of myelin on the axonal arbor of individual neocortical neurons have been difficult to undertake with traditional methods. Golgi labeling of neurons halts where the axon first enters a myelin sheath (*Peters and Proskauer, 1980*; *DeFelipe et al., 1986*; *Fairén et al., 1977*), and bulk labeling of myelin reveals a laminar patterned thicket (*Nieuwenhuys, 2013*), rather than individual neuronal arbors.

Recent advances in large volume, high resolution neuroanatomy methods (*Briggman and Bock, 2012*; *Micheva and Smith, 2007*; *Economo et al., 2016*) now permit cortical myelin to be directly visualized on the axonal arbors of multiple individual identified neurons for all axons within the imaged volume. Myelin tracing in large EM datasets has revealed that myelination of the axons of pyramidal neurons in cortex is distinct from the known myelination patterns in subcortical white matter, and also differs between cortical layers (*Tomassy et al., 2014*). About 10–20% of neocortical neurons are inhibitory and some of these neurons may also have myelinated axons. For example, labeling of individual neocortical neurons during in vivo electrophysiological recording, followed by light-level reconstruction and electron microscopy (EM) of selected sub-arbors, has revealed that inhibitory neurons may be myelinated (*DeFelipe et al., 1986*; *Somogyi et al., 1983*; *Kawaguchi and Kubota, 1998*; *Kisvarday et al., 1987*; *Thomson et al., 1996*; *Tamás et al., 1997*). Complementary immunohistochemical studies have shown that parvalbumin-positive (i.e., putative inhibitory [*Rudy et al., 2011*]) axons may be myelinated (*McGee et al., 2005*; *Chung and Deisseroth, 2013*). Given that cortical interneuron types are implicated in distinct circuit functions

(*Kepecs and Fishell, 2014*) and have distinct behavioral correlates (*Kvitsiani et al., 2013*), these results raise several important questions regarding the organization of neocortical neuronal networks: how widespread is inhibitory axon myelination? Is it specific to certain interneuron subtypes? Does inhibitory axon myelin differ from excitatory axon myelin at the molecular level?

Here we combine analyses of light-microscopy-based array tomography (AT) data (*Micheva and Smith, 2007*; *Collman et al., 2015*) with a publicly hosted EM dataset (*Burns et al., 2013*; *Martone et al., 2002*) generated using a custom transmission EM camera array (TEMCA) (*Bock et al., 2011*), to obtain an integrated view of the myelination of inhibitory neurons in cortex. We find that an unexpectedly large proportion of myelinated axons in cortex arise from a specific subtype of inhibitory neuron: the parvalbumin-positive (PV+) basket cell. These PV+ axons are especially prominent in the middle cortical layers. They constitute about half of all myelinated axons in layers 2/3, and a quarter of myelinated axons in layer 4. We also find that the inhibitory myelinated axons in cortical gray matter differ from the excitatory myelinated axons in many respects, such as their structural organization, cytoskeletal content, and myelin protein composition. Taken together, the abundance and cell-type selectivity of neocortical myelinated inhibitory axons described here raise fundamental questions about the role of myelination both in normal function and in disease.

## Results

We applied two microscopic approaches to determine the neurotransmitter content of cortical myelinated axons. Array tomography (AT) is based on digital reconstruction of images acquired from arrays of serial ultrathin sections (70 nm) attached to coverslips and imaged with different modalities, such as immunofluorescence and scanning electron microscopy (SEM). The use of ultrathin sections allows the unambiguous light level identification of individual myelinated axons even in regions with high myelin density (e.g., subcortical white matter), while the possibility of applying multiple immunofluorescent markers (10 or more) enables the molecular characterization of these axons. Conjugate SEM further reveals the underlying ultrastructure. TEMCA-based volume electron microscopy uses an array of high-speed cameras to efficiently image large numbers of serial ultrathin (40–50 nm) sections. The resulting image volumes are of sufficient size and resolution that the densely packed and intertwined axons and dendrites of the brain neuropil can be traced over relatively long distances, and the synaptic connections between them can be discerned. With this form of EM, all membranes and synapses are labeled, enabling neuronal structure and connectivity to be sampled in an unbiased fashion. Additional evidence was also obtained using immunohistochemical labeling of brain slices from transgenic mice expressing fluorophores in salient inhibitory interneuron subtypes. The EM and AT studies were initiated and conducted independently in two different laboratories and data were pooled once the complementary nature of the findings were ascertained (Materials and methods).

### Array tomography shows that a large fraction of myelinated axons in upper layers of cortex contain the inhibitory neurotransmitter GABA

Immunofluorescence AT is particularly well suited for the study of myelinated axons in cortex. The abundance of myelin proteins, such as myelin basic protein (MBP), and the size of myelinated axons (>200 nm in diameter) make them an ideal target for immunofluorescence detection (*Figure 1A*, *Figure 1—figure supplement 1*, *Video 1*). The ultrathin AT sections provide easy access of the antibodies against MBP and do not require lipid extraction. Conjugate light-electron AT confirms that immunofluorescence for MBP precisely outlines the myelinated sheaths of axons (*Figure 1B*, *Figure 1—figure supplement 1B*, *Figure 1—figure supplement 2A*). Overall, the MBP immunolabel is of high quality as evidenced by the consistency of the signal from section to section (*Figure 1—figure supplement 2B*), and the low background (mean gray value of 248 ± 6 a.u. on resin and 279 ± 2 a.u. on nuclei, vs 5754 ± 226 a.u. on myelin, mean and standard error). In addition, the use of ultrathin sections ensures that antigens can be detected equally well within myelinated and unmyelinated portions of the axon. For example, in our AT experiments GABA and PV immunostaining within axons is not affected by myelination (*Figures 1*, 3), in contrast to preembedding immunocytochemistry where myelinated portions of axons are much more weakly stained (*Pawelzik et al., 2002*).

As expected, AT detects the presence of myelinated axons in all neocortical layers. Quantification of the number of MBP profiles in single sections shows that their numbers dramatically increase with

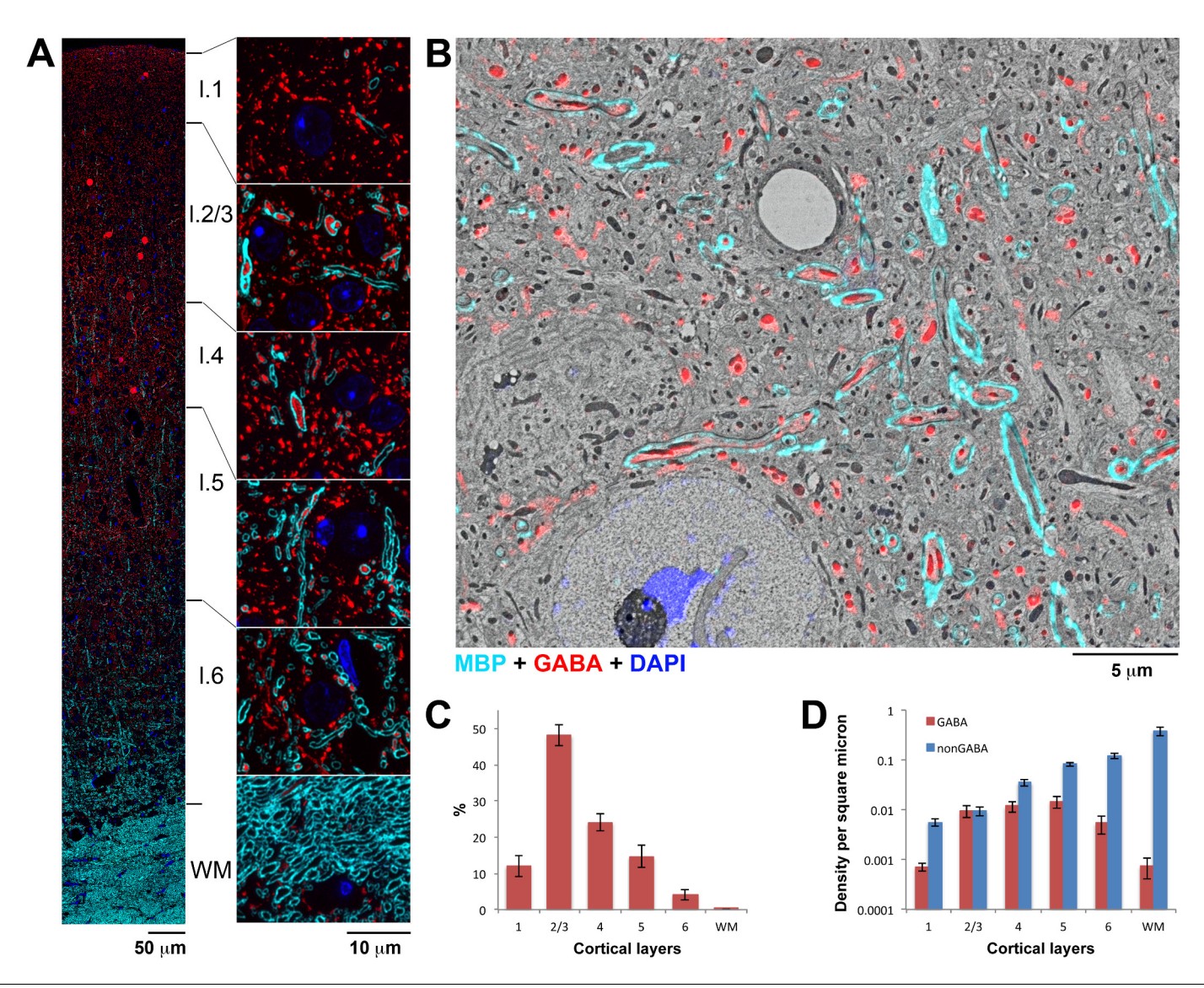

**Figure 1.** A large fraction of myelinated axons in upper layers of cortex contain GABA. (**A**) An ultrathin section (70 nm) through the mouse somatosensory cortex immunolabeled for MBP (cyan) and GABA (red). Nuclei are stained with DAPI (blue). Smaller regions from each layer are shown at higher magnification to the right. (**B**) Scanning electron micrograph from layer 5 of the mouse somatosensory cortex overlaid with the corresponding immunofluorescence for MBP (cyan) and GABA (red), and the DAPI signal (blue). (**C**) Proportion of myelinated axonal profiles containing GABA in the cortical layers of mouse somatosensory cortex. (**D**) Density of GABA and non-GABA myelinated axons in mouse somatosensory cortex (y-axis is in logarithmic scale to accommodate the large range of myelinated axon densities along the cortical depth). Mean from 3 animals and standard errors are shown in **C** and **D** (number of axonal profiles analyzed per animal was 11103, 12,657 and 7540, respectively).

The following source data and figure supplements are available for figure 1:

**Source data 1.** Data values underlying *Figure 1*.

**Figure supplement 1.** Antibodies used in this study labeled the expected structures as assessed by IF volume reconstruction and SEM imaging.

**Figure supplement 2.** MBP antibody performance.

cortical depth (*Figure 1D*), consistent with previous quantitative myeloarchitectonic studies in non-rodent species (*Nieuwenhuys, 2013*; *Braitenberg, 1962*; *Hopf, 1966*). Immunostaining for the inhibitory neurotransmitter GABA reveals that a surprisingly large number of myelinated axons throughout cortex contain GABA, including more than 45% of myelinated axons in layers 2/3 (48.1 ± 3.2%, mean ± standard error, *Figure 1C*). Layers 4 and 5 exhibit similar numbers of GABA myelinated axons as layers 2/3 (*Figure 1D*), but the deeper layers contain larger numbers of non-GABA myelinated axons, presumably of projecting pyramidal cells. GABA myelinated axons are relatively sparse in layers 1, 6, and only occasionally seen in subcortical white matter.

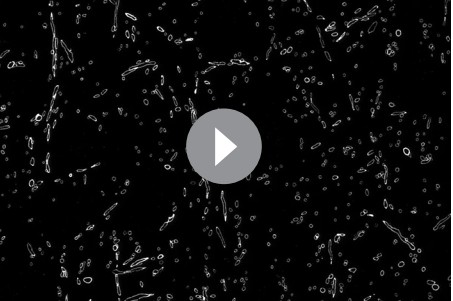

**Video 1.** Serial sections through mouse cortex immunostained for MBP. Stack of raw images from 59 serial sections (70 nm each) from cortical layer 5.

## Volume electron microscopy reveals that a large fraction of myelinated axons in layer 2/3 form symmetrical (presumed inhibitory) synapses

In parallel with the array tomography work, and blind to its results, myelinated axons were traced within a previously acquired (*Bock et al., 2011*), publicly available (*Burns et al., 2013*; *Martone et al., 2002*) volume electron microscopy dataset from the upper layers of mouse visual cortex V1. This dataset was of sufficient size and resolution that both myelinated and unmyelinated axons could be traced over hundreds of microns, allowing them to be categorized on the basis of their synaptic ultrastructure as inhibitory or excitatory. A section from near the middle of the EM volume was selected, and all the myelin profiles from the center portion of this section were annotated (*Figure 2A*). Each of these myelinated 'seeds' (e.g. *Figure 2B,E*) was used as a starting point for further tracing of the myelinated axon through the EM volume. Along their length, a majority of the traced axons unmyelinated and made a number of synapses (*Figure 2C,F*). Axons were classified according to the appearance of the pre- and postsynaptic densities of the synapses they made, using the classical definitions for asymmetrical (excitatory) and symmetrical (inhibitory) synapses (*Colonnier, 1968*; *Peters et al., 1991*) (*Figure 2D,G*). Of the 231 myelinated 'seeds', 73 were classified as excitatory (31.6%; *Figure 2—figure supplement 1*), 106 as inhibitory (45.9%; *Figure 2—figure supplement 2*), and 52 (22.5%; *Figure 2—figure supplement 3*) could not be categorized due to an absence of synapses within the volume. This analysis shows that inhibitory myelinated axons are abundant in cortical layer 2/3, comprising at least 45% of myelinated axons in the mouse visual cortex.

## Nearly all myelinated GABA axons are parvalbumin-positive

Cortical GABA neurons are a highly diverse population (*Ascoli et al., 2008*; *Markram et al., 2004*; *Xu et al., 2010*). About half of them contain parvalbumin (PV), a calcium buffering protein. Array tomography (*Figure 3A,B* and *Figure 3—figure supplement 1*) revealed that nearly all GABA myelinated axons (97.9 ± 0.8%, mean ± st.error from 3 animals) are immunopositive for parvalbumin and therefore originate from PV interneurons. Correspondingly, in the examined cortical AT volumes (*Figure 3C–F*), examples of PV and GABA-positive cell bodies forming myelinated axons are seen in layers 2/3, 4 and 5. Blind to the AT results, we conducted independent experiments in three mice expressing the fluorescent protein tdTomato driven by Cre expression in PV neurons (*Pvalb-ires-Cre; Ai9*) (*Taniguchi et al., 2011*). Confocal imaging of 40 µm thick Vibratome slices from these mice immunostained with MBP corroborated the AT findings by showing that about half (53 ± 12%) of the myelinated axons in layer 2/3 of somatosensory cortex contain tdTomato and, therefore, PV (*Figure 4A–D*). Similar experiments were also performed on mice expressing tdTomato from SOM-Cre or VIP-Cre driver lines (*Sst-ires-Cre;Ai9*, and *Vip-ires-Cre;Ai9*) (*Taniguchi et al., 2011*; *Madisen et al., 2010*). After careful inspection of selected subareas, no myelinated axons containing reporter for the non-PV cortical interneurons vasoactive intestinal polypeptide (VIP; *Figure 4E–G*) could be observed. A small fraction (4.4%) of somatostatin (SOM)-positive myelinated axons were

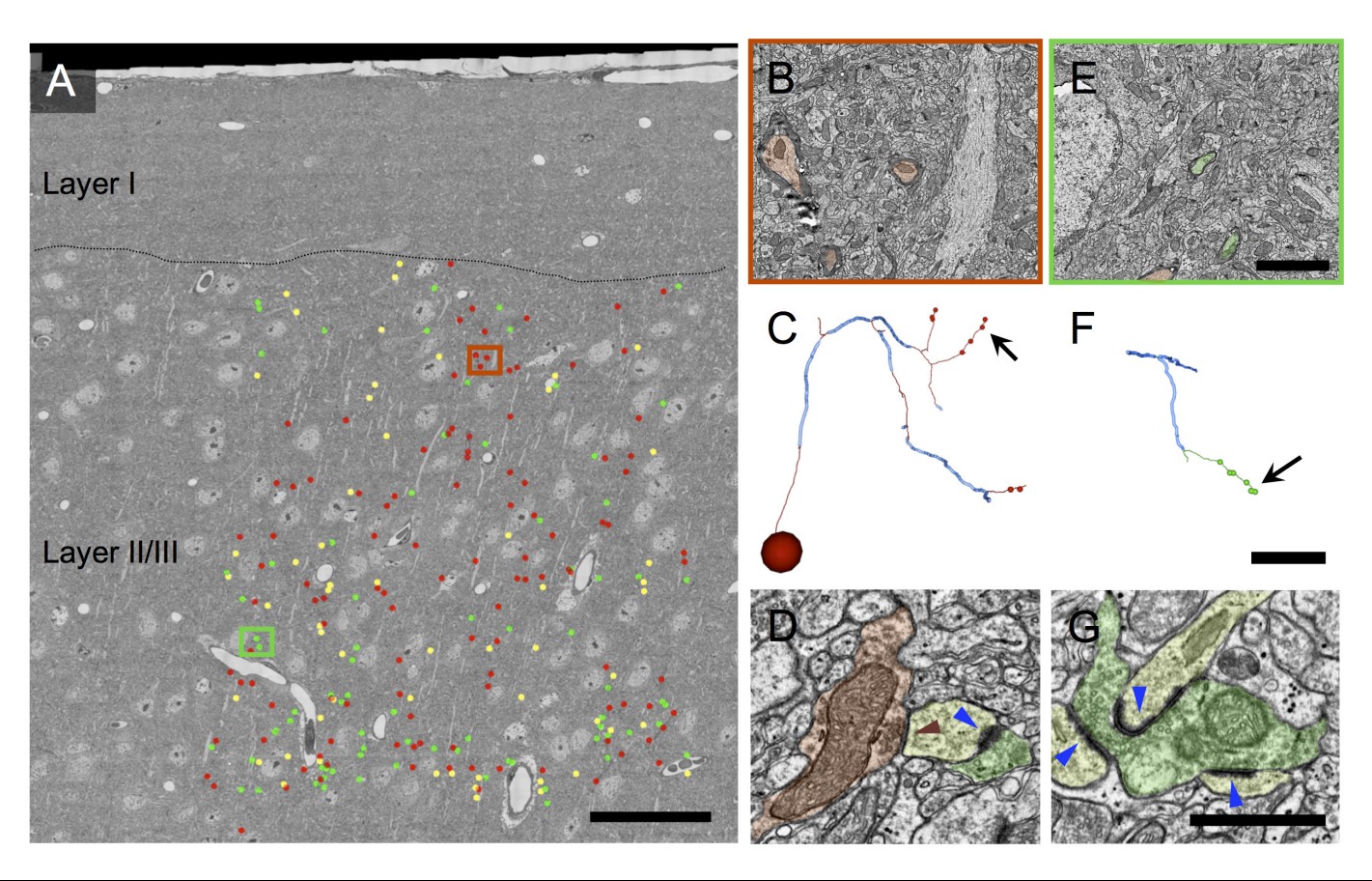

**Figure 2.** Volume electron microscopy shows a large fraction of myelinated axons in layer 2/3 of cortex are inhibitory. (A) EM mosaic from pia through lower layer 2/3, overlaid with the locations of myelinated axon profiles used as tracing seeds. Each seed is color-coded by the type of synapses the axon was determined to make after exiting its myelin sheath elsewhere in the volume (red: symmetric; green: asymmetric; yellow, no synapses in the EM volume). Red and green rectangles indicate areas of detail in B and E, respectively. (B) Detail view of area in red rectangle in A. Three inhibitory myelinated axon profiles (false colored red) are shown. (C) A reconstructed axon arbor arising from the myelinated tracing seed at the center of B. Thick blue segments indicate myelinated internodal regions; thin red segments represent unmyelinated axon. Segment diameters are schematic. This axon was traced to its originating soma (sphere in lower left) and to 8 synapses (small red dots). The arrow indicates the location on this axon's arbor of the symmetric synapse shown in D. Note that axon arbors were traced only until their synapses could be reliably categorized as symmetric or asymmetric; therefore the arbor shown here is a small subset of the full axonal arbor arising from this inhibitory neuron. (D) An unmyelinated axon profile (red) makes a symmetric synapse (red triangle) onto a postsynaptic spine (yellow). This spine also receives an asymmetric synapse (blue triangle) from an excitatory axon (green). (E) Detail view of area in green rectangle in A, showing two excitatory myelinated axon profiles (false colored green). (F) A reconstructed axon arbor arising from the myelinated tracing seed at the center of E. Conventions as in C, except unmyelinated axon segments are rendered in green. (G) An unmyelinated axon profile (green) makes three asymmetric synapses (blue triangles) onto three different spines. Scale bar in A, 50 µm; E, 3 µm (also applies to B); F, 20 µm (also applies to C); G, 1 µm (also applies to D).

The following figure supplements are available for figure 2:

**Figure supplement 1.** Distribution of myelin (cyan) on axon fragments (green) categorized as excitatory.

**Figure supplement 2.** Distribution of myelin (cyan) on axon fragments (red) categorized as inhibitory.

**Figure supplement 3.** Distribution of myelin (cyan) on axon fragments (yellow) that could not be categorized, due to an absence of synapses in the EM-imaged volume.

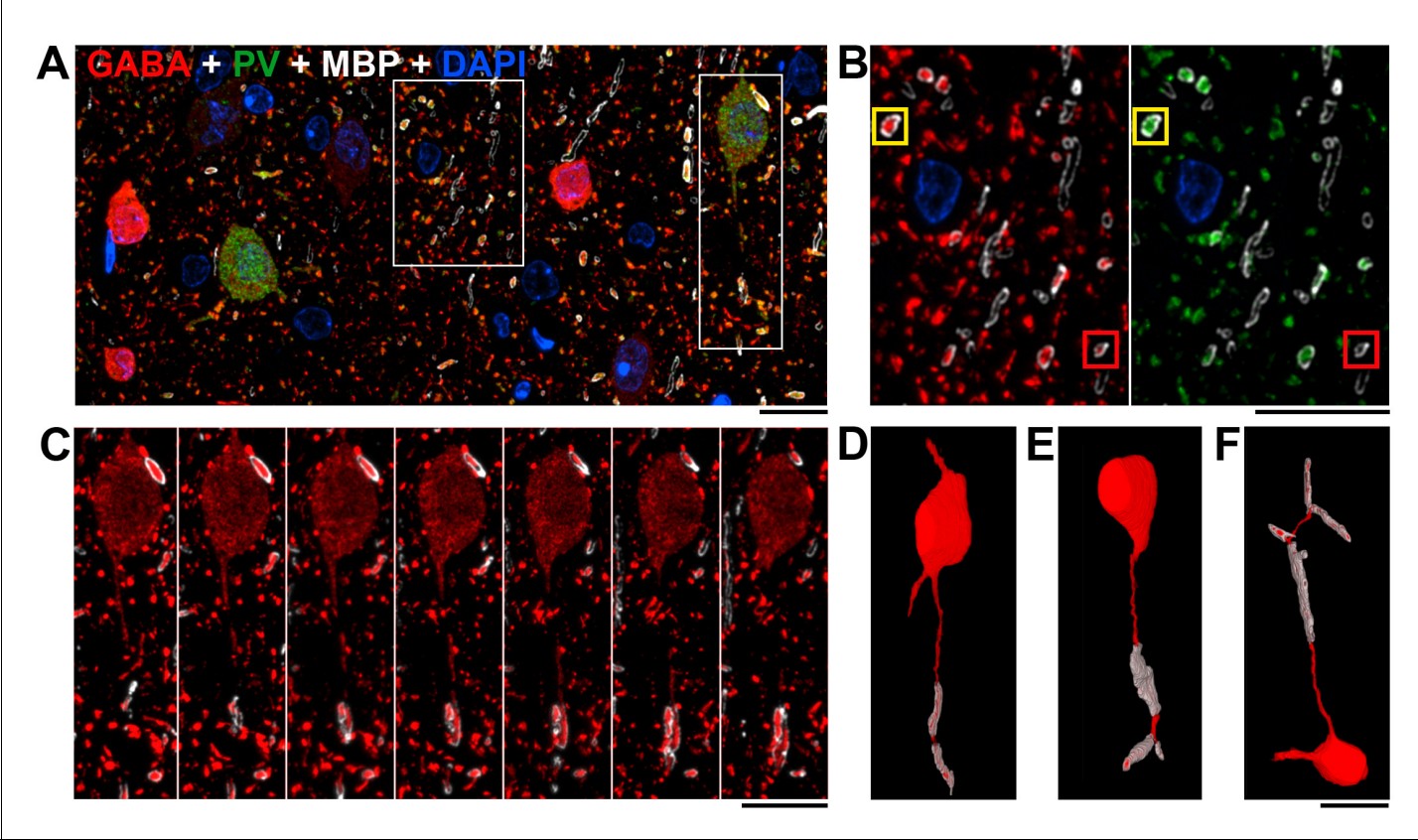

**Figure 3.** Nearly all myelinated GABA axons are parvalbumin-positive. (**A**) A single section (70 nm) from layer 2/3 of mouse cortex, immunolabeled for MBP (white), GABA (red) and PV (green); nuclei are stained with DAPI (blue). Note that the PV-containing neurons have weaker GABA immunoreactivity. (**B**) The central box from **A** shown at a higher magnification. The great majority (97.9 ± 0.8%) of GABA immunopositive myelinated axons (red, left panel) also contain PV (green, right panel); the yellow box marks an example of such an axon (see main text for quantification). Occasionally, a GABA myelinated axon does not show detectable PV immunofluorescence as shown in the red box. (**C**) Serial sections through the parvalbumin containing neuron boxed in **A**, **C** showing its myelinated axon. (**D**) Volume reconstruction of the neuron in **C**. (**E**, **F**) Volume reconstructions of PV interneurons with myelinated axons from layers 4 and 5. **A–F**, Scale bar, 10 μm.

The following figure supplement is available for figure 3:

**Figure supplement 1.** GABA and PV immunoreactivity of myelinated axons.

observed (*Figure 4H–J*); however, the transgenic reporter line used was recently found to have a small but consistent false-positive expression pattern, wherein up to 10% of Cre-expressing neurons are SOM-negative and PV-positive when examined by immunohistochemistry (*Hu et al., 2013*). At least some of the SOM-positive myelinated axon profiles we observe therefore may be false positives which actually express PV, not SOM; and some may be true positives, consistent with the AT data, which show that ~2% of GABA myelinated profiles are PV negative. Regardless, the data obtained from the three Cre-lines support the conclusion that the great preponderance of myelinated inhibitory axon profiles in layer 2/3 arise from parvalbumin-positive axons.

## Myelinated inhibitory axons most likely belong to cortical basket cells

Among the inhibitory neurons of cortex, parvalbumin is found in basket cells, chandelier cells (*Markram et al., 2004*; *Kawaguchi and Kubota, 1997*; *DeFelipe et al., 1989*), and in a sparse subpopulation of long-range inhibitory axons arising from basal forebrain (*Freund and Gulyás, 1991*; *Caputi et al., 2013*). Because these different cell types are known to have distinct axonal morphology and targets, we traced the postsynaptic targets of inhibitory myelinated axons in layer 2/3 from the volume EM data set in an effort to understand their originating cell type (*Figure 5*). The

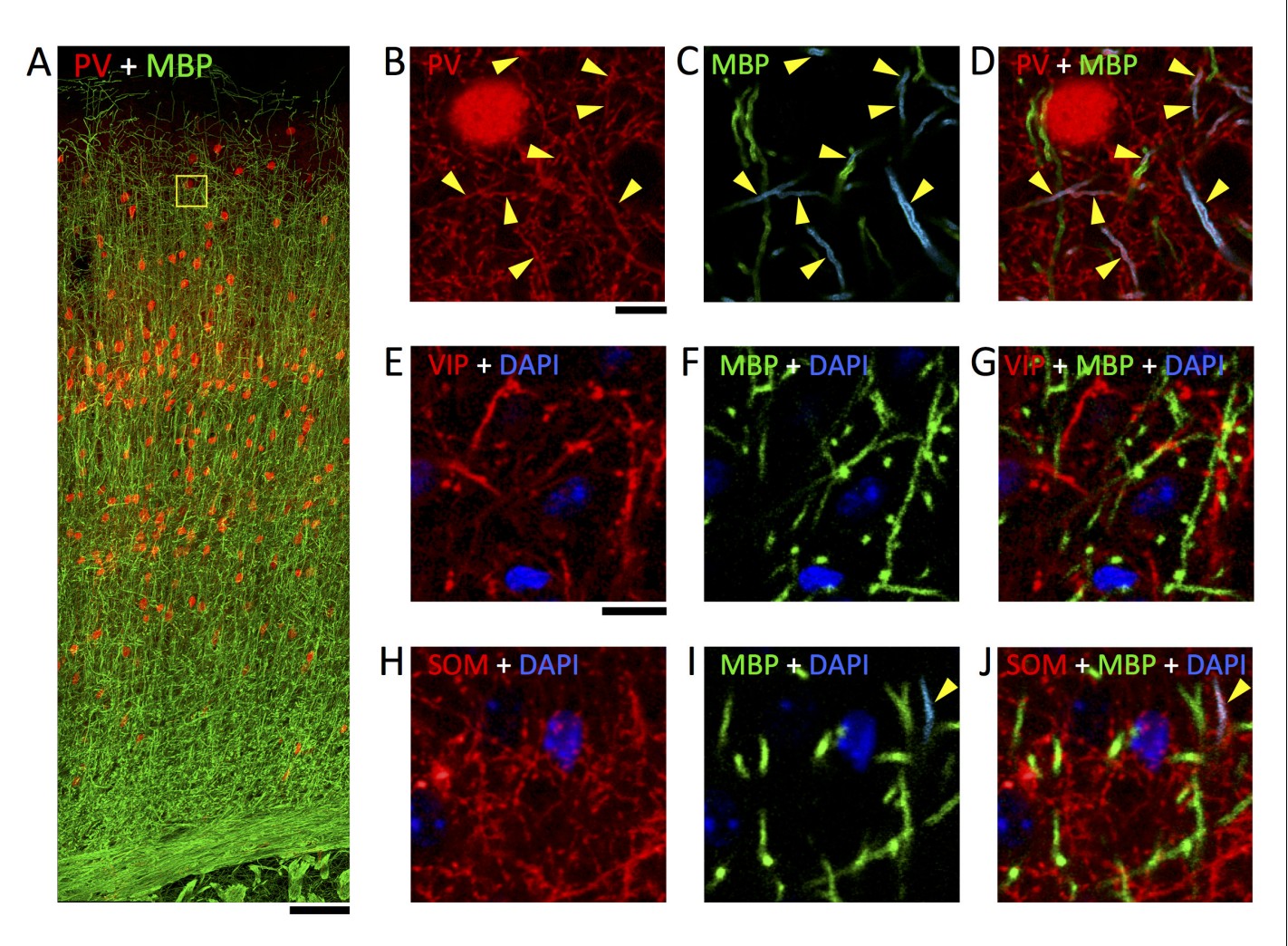

**Figure 4.** Immunofluorescent labeling of MBP shows half of myelin profiles ensheath PV-positive axons but no VIP- and very few SOM-expressing axons. (A) Maximum intensity projection of a stack of confocal image mosaics, spanning pia (top) to white matter (bottom) of somatosensory cortex, revealing the laminar distribution of PV-expressing neurons (red) and immunolabeled myelin (green). Yellow square in layer 2/3 indicates the location of the field of view portrayed in panels B–D. (B–D) Detail view of a representative subarea from a single section in layer 2/3. About half of the myelinated axons are PV-positive (yellow triangles). B shows the red channel (PV) only; C shows the green channel (myelin) only, and myelinated profiles containing PV-positive axons are false-colored cyan; and D overlays panels B and C. (E–G) Representative confocal images from layer 2/3 of somatosensory cortex, showing VIP-expressing neurites (red), immunolabeled myelin (green), and DAPI-stained nuclei (blue). E shows only the red (VIP) and blue (DAPI) channels; F shows only the green (myelin) and blue (DAPI) channels; and G overlays panels E and F. (H–J) Representative confocal images from layer 2/3 of somatosensory cortex, showing SOM-expressing neurites (red), immunolabeled myelin (green), and DAPI-stained nuclei (blue). H shows only the red (SOM) and blue (DAPI) channels; I shows only the green (myelin) and blue (DAPI) channels, and myelinated profiles containing SOM-positive axons are false-colored cyan; and J overlays panels H and I. No myelinated axon profiles are positive for VIP; 4.4% are positive for SOM. Scale bar in A, 100 μm; B, 10 μm (also applies to C–D); E, 10 μm (also applies to H–J).

postsynaptic targets were identified as inhibitory or excitatory according to previously described ultrastructural criteria (*Bock et al., 2011*) (Materials and methods). Briefly, postsynaptic dendrites were classified as inhibitory if they were largely aspinous and densely coated with asymmetric (excitatory) synapses (e.g. *Figure 6E–F*), while they were classified as excitatory if they were spiny and received fewer shaft excitatory synapses. Excitatory somata were distinguished from inhibitory somata based on cell shape, the presence of a large apical dendrite, and the absence of asymmetric contacts on the soma. For each synapse, the postsynaptic compartmental location and class

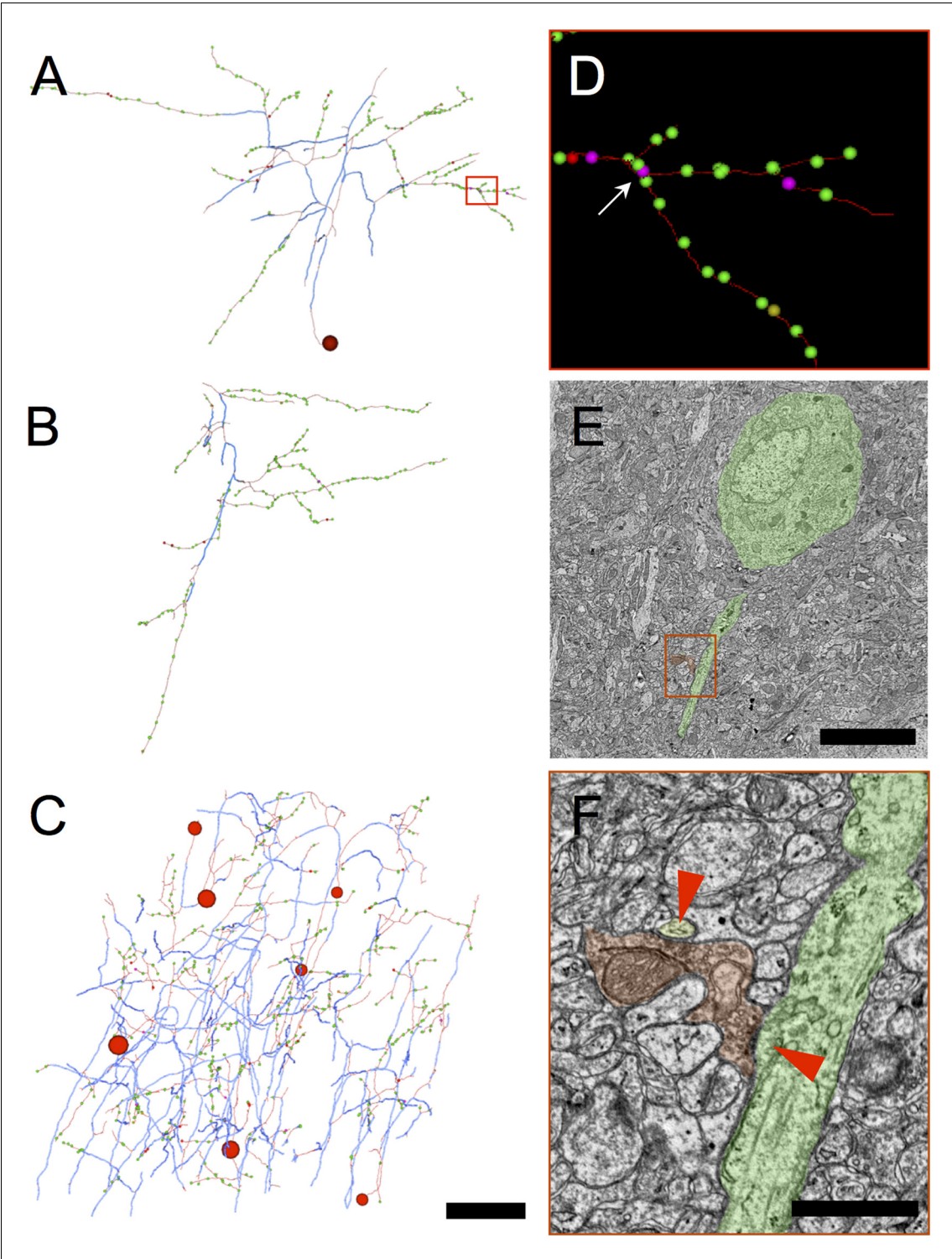

**Figure 5.** Myelinated inhibitory axons rarely make synapses upon other axons and are therefore unlikely to belong to chandelier cells. (A) An inhibitory axon traced from a myelinated seed to completion within the EM volume. The proximal axonal arbor receives the preponderance of myelin (thick blue arbor segments), whereas the distal arbor makes most of the synapses. The red rectangle outlines the sub-arbor shown in D. In (A–C), the large red spheres indicate the position of the soma. Synapses are represented by dots, color-coded according to postsynaptic target class: green (excitatory), magenta (excitatory axon), yellow (unclassifiable), and red (inhibitory). (B) A second inhibitory axon traced to completion. In this case the EM volume boundaries were reached before reaching the soma. A myelinated core portion of the arbor can be discerned, with most synapses at the periphery. (C) All the inhibitory axon fragments traced from myelinated seeds. Overall, only 2% of synapses made by myelinated inhibitory axons are onto excitatory axons (magenta spheres); no synapses onto inhibitory axons were observed. (D) Detail view of area shown in red rectangle in **A** showing one of the rare

*Figure 5 continued on next page*

*Figure 5 continued*

synapses made by myelinated inhibitory neurons onto the proximal axon of a pyramidal cell (arrow). (E) EM image of a section intersecting the pyramidal cell body and its proximal axon (false-colored green), postsynaptic to the synapse indicated by the arrow in D (false-colored red). The red rectangle indicates the area of detail shown in F. (F) A magnified view of the synapse shown in E. A second postsynaptic target, the spine neck of a dendrite arising from a different pyramidal cell is false-colored in yellow. Symmetrical synapse locations indicated by red triangles. Scale bar in A–C, 50 μm; E, 4 μm; F, 1 μm.

(excitatory or inhibitory) was tabulated (*Table 1*), and postsynaptic class was used to color-code the synapses along the reconstructed axonal fragments (*Figures 5–6*).

The observed distribution of postsynaptic targets of myelinated inhibitory axons was consistent with their originating from local axonal arbors of cortical basket cells, and inconsistent with their arising from either chandelier cells or long-range inhibitory afferents to cortex. Basket cells and chandelier cells can readily be distinguished by their morphology and synaptic targets. When the axons from multiple basket cells are labeled and examined at the light level, they form 'baskets' of synapses around neuronal somata, hence the name (*Kisvárday, 1992*). Chandelier cells have an axonal arbor which resembles a chandelier, with vertically oriented axonal cartridges that almost exclusively form synapses onto the axon initial segments of pyramidal neurons (*Somogyi et al., 1982*). Less than 2% of the inhibitory myelinated axons in our volume electron microscopy dataset were found to make contacts on axon initial segments (*Figure 5*; *Table 1*). In addition, the few axonal arbors that were seen contacting axon initial segments did not have the characteristic appearance of chandelier cell axons and made single contacts with axonal initial segments, instead of several (typically 3–5 for chandelier cells) synapses in a row (*Figure 5A,B*). Therefore we can exclude the possibility that the myelinated inhibitory axons arise from chandelier cells. The great preponderance (93%) of postsynaptic targets of myelinated inhibitory axons in layer 2/3 of cortex were excitatory, likewise eliminating the possibility that these axons originated from the arbors of the known parvalbumin-positive long-range inhibitory afferents to cortex, all of which selectively target inhibitory neurons (*Freund and Gulyás, 1991*; *Gritti et al., 2003*; *Henny and Jones, 2008*).

The remaining possibility, that myelinated inhibitory axons arise from the arbors of local parvalbumin-positive basket cells, was well supported by the distribution of their postsynaptic targets. Although neocortical basket cells are best known for their synapses with cell bodies, a large (but variable) fraction of their synapses are with dendritic shafts and spines (*Somogyi et al., 1983*; *Kawaguchi and Kubota, 1998*; *Kisvarday et al., 1987*; *Tamás et al., 1997*). Consistent with past surveys of basket cell synaptic targets, we found that 13% of the postsynaptic targets of the myelinated inhibitory axons were with somata, 49% were onto excitatory dendritic shafts, and 30% were onto dendritic spines (*Figure 6*; *Table 1*). Thus, we conclude that the majority of myelinated inhibitory axons belong to the intracortical axons of parvalbumin containing basket cells.

## Inhibitory neurons exhibit a distinct pattern of myelination

Both our volume electron microscopy and array tomography data show that the axon of a parvalbumin interneuron becomes myelinated soon after exiting the cell body (usually within 20–50 μm), consistent with myelination commencing shortly after the axon initial segment. All of the axonal arbors that were traced back to the cell body in our volume EM dataset (n=8) had the same general appearance: a central core of partially myelinated axons (whose unmyelinated stretches rarely made synapses), from which emerged unmyelinated branches forming numerous synapses (*Figures 2C*, *5A,C*). This is similar to the pattern reported for filled basket cells in both macaque somatosensory cortex (*DeFelipe et al., 1986*) and cat visual cortex (*Somogyi et al., 1983*).

Further analysis revealed marked differences in the myelination of GABA and nonGABA axons. We used array tomography and MBP immunostaining to quantify the lengths of the nodes of Ranvier, where the ion channels needed for nerve impulse transmission are concentrated (*Figure 7A–B*). Only gaps in the axonal myelin sheath of less than 4 μm length were considered to be nodes. Throughout the cortical layers, the nodes of GABA axons were shorter than for non-GABA axons ($0.98 \pm 0.07$ μm vs. $1.51 \pm 0.05$ μm, mean ± standard error, $p < 0.0001$, n=71 GABA and 209 non-GABA nodes of Ranvier, Mann-Whitney U Test; *Figure 7C*). Their internodes, the myelinated portion between two nodes of Ranvier, were also found to be shorter ($22.54 \pm 2.11$ μm vs $29.29 \pm 2.02$ μm

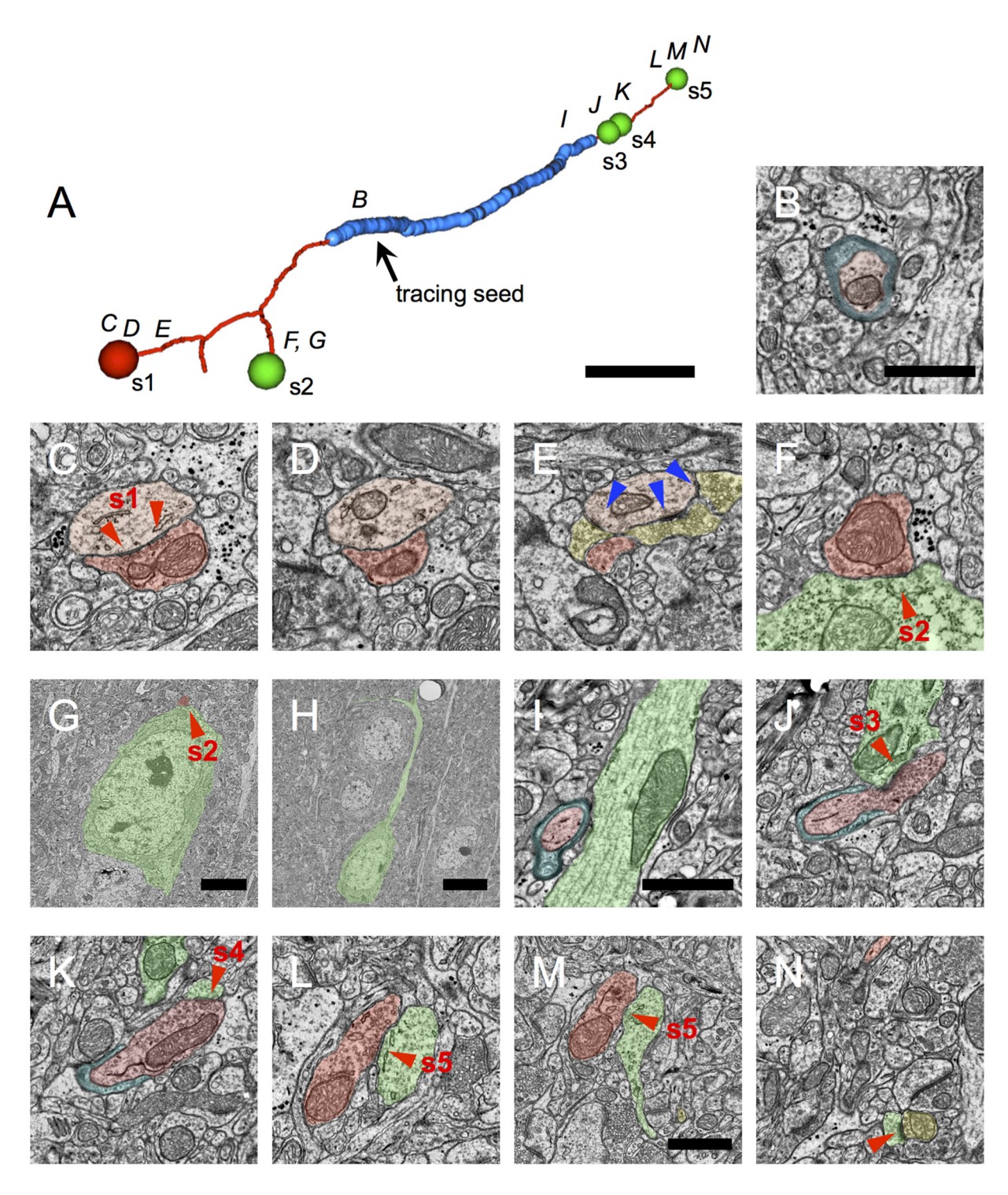

**Figure 6.** Myelinated inhibitory axons target dendritic shafts, spines and neuronal cell bodies. (A) A representative inhibitory axon fragment traced from a myelinated seed profile (arrow). Blue indicates the extent of myelin along the fragment; thin red axon segments are unmyelinated. The spheres indicate the location of symmetric synapses, and are color-coded red for inhibitory postsynaptic targets (synapse 1) and green for excitatory postsynaptic targets (synapses 2–5). These five symmetric synapses were sufficient to categorize the axon as inhibitory. The letters above the fragment correspond to the approximate locations of later panels. (B) The myelinated profile used as a tracing seed for the axon fragment in A. Myelin is false-colored blue; the axon is false colored in red. (C–N) EM micrographs through synapses 1–5. The presynaptic axon is false-colored in dark red; the postsynaptic inhibitory dendrite in light red; the postsynaptic excitatory targets in green; and other excitatory boutons converging onto the same

*Figure 6 continued on next page*

*Figure 6 continued*

targets as the myelinated inhibitory axon, in yellow. (C) Synapse 1 contacts an inhibitory dendrite, which is also shown in (D–E). In E, the dendrite is densely coated with excitatory axonal boutons making asymmetric synapses (blue triangles), a hallmark of inhibitory dendrites. (F) Cross-section through synapse 2. (G) Larger field of view of the section in F, showing that the postsynaptic target is a neuronal cell body. (H) A nearby section through the same neuron as in G, reveals it to be a pyramidal cell with a prominent apical dendrite. (I) The myelinated axon approaching an apical dendrite. (J) Synapse 3 is formed immediately after the axon unmyelinates and contacts the same apical dendrite shown in I. (K) The same bouton participating in synapse 3 forms synapse 4 with a spine arising from the same apical dendrite shown in I and J. (L) Synapse 5 targets the shaft of a dendrite, identified as excitatory by the presence of spines on near-by sections (e.g. M–N). (M) Synapse 5 can still be seen, and a spine neck arises from the postsynaptic dendrite. (N) The spine neck in M forms a small spine head, receiving an asymmetric synapse (blue triangle) from an excitatory bouton from a different axon. Scale bar in A, ~1 µm; B, 1 µm (also applies to C–F); G, 4 µm; H, 9 µm; I, 1 µm (also applies to J–L); M, 1 µm (also applies to N).

for non-GABA axons, mean ± standard error, p=0.032, n=23 GABA axons and 32 non-GABA axons, Mann-Whitney U Test).

## GABA and non-GABA myelin differ in protein composition, but not thickness

Myelin thickness is thought to be directly correlated to axon diameter (constant 'g-ratio'), based mostly on studies of the peripheral nervous system (*Fraher and Dockery, 1998*). However, we found no such correlation within mouse cerebral cortex. Using a deconvolved array tomography dataset with MBP and GABA immunofluorescence, we found that the average myelin thickness was 0.13 ± 0.002 µm for both GABA and non-GABA axons (mean ± standard error, n=163 GABA and 238 non-GABA axons from layers 4 and 5, p=0.88, Mann-Whitney U Test) and it did not correlate with axon thickness (R=0.008; n=163 GABA and 238 non-GABA axons from layers 4 and 5). Greater axon diameters not accompanied by changes in myelin thickness result in the g-ratio increasing as a function of axon diameter (*Figure 8B*). GABA axons are thicker than non-GABA axons (0.54 ± 0.001 µm vs. 0.45 ± 0.001 µm, n=163 GABA and 238 non-GABA axons, p<0.0001, Mann-Whitney U Test; and *Figure 8A*), thus their average g-ratio is higher (0.66 ± 0.006 vs. 0.61 ± 0.006, n=163 GABA and 238 non-GABA axons, p<0.0001, Mann-Whitney U Test).

Next we compared the protein composition of the myelin of GABA and non-GABA axons. The two major proteins in myelin are MBP and proteolipid protein (PLP), which together constitute about 80% of all proteins in myelin (*Baumann and Pham-Dinh, 2001*; *Rosetti and Maggio, 2007*). While the average PLP immunofluorescence was very similar in both types of axons (2229 ± 60 a.u. vs 2282

**Table 1.** The distribution of the postsynaptic targets.

| Postsynaptic target | Completely traced axons | | Partially traced axons | | All | |
|---|---|---|---|---|---|---|
| | count | % | count | % | count | % |
| Excitatory somata | 60* | 15.0 | 46[†] | 10.5 | 106 | 12.6 |
| Excitatory dendritic shafts | 184 | 45.9 | 225 | 51.4 | 409 | 48.7 |
| Excitatory dendritic spines | 119 | 29.7 | 134 | 30.6 | 253 | 30.2 |
| Excitatory axons | 7 | 1.7 | 8 | 1.8 | 15 | 1.8 |
| *All excitatory* | *370* | *92.3* | *413* | *94.3* | *783* | *93.3* |
| Inhibitory dendritic shafts | 17 | 4.2 | 20 | 4.6 | 37 | 4.4 |
| Inhibitory dendritic spines | 0 | 0 | 0 | 0 | 0 | 0 |
| Inhibitory axons | | | 1 | 0.2 | 1 | 0.1 |
| *All inhibitory* | *17* | *4.2* | *21* | *4.8* | *38* | *4.5* |
| Uncategorized dendritic spines | 14 | 3.5 | 4 | 0.9 | 18 | 2.1 |
| Total | 401 | | 438 | | 839 | |

* Including 10 synapses onto somatic spines;

[†] Including 8 synapses on somatic spines

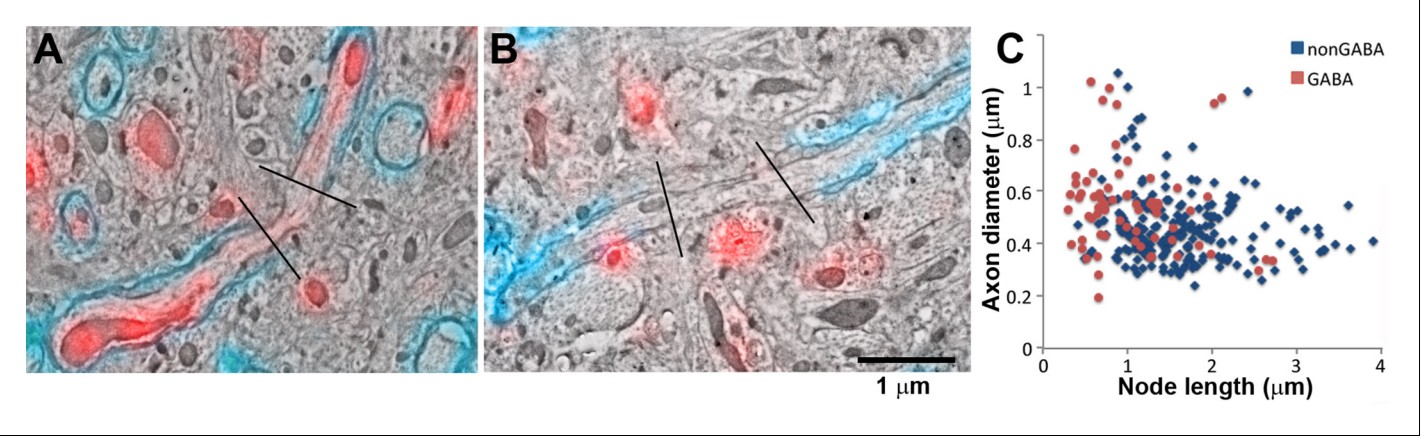

**Figure 7.** GABA axons have shorter nodes of Ranvier. (A, B) SEM images of nodes of Ranvier (black lines: node boundaries) of a GABA (A) and non-GABA myelinated axon (B), immunolabeled with MBP (cyan) and GABA (red). (C) Comparison of the lengths of the nodes of Ranvier of cortical myelinated axons.

The following source data is available for figure 7:

**Source data 1.** Data values and statistics underlying *Figure 7*.

± 40 a.u., for GABA and non-GABA axons respectively; mean ± standard error, n=254 GABA axons and 489 non-GABA, p=0.2713, Mann-Whitney U Test, *Figure 8D*), GABA myelinated axons had almost 20% higher MBP immunofluorescence (5263 ± 117 a.u. vs. 4486 ± 63 a.u., mean ± standard error, n=254 GABA axons and 489 non-GABA, p<0.0001, Mann-Whitney U Test). This difference in MBP content was not related to the thickness of the axons (*Figure 8C*). The higher MBP immunofluorescence of the myelin sheath of inhibitory axons was confirmed in experiments in two more animals (A1: 3024 ± 100 a.u. vs. 2634 ± 86 a.u., mean ± standard error, n=322 PV axons and 314 non-PV from layers 2 to 5, p=0.0114, Mann-Whitney U Test; A2: 3038 ± 117 a.u. vs. 2585 ± 120 a.u., mean ± standard error, n=233 PV axons and 119 non-PV from layer 4, p=0.0455, Mann-Whitney U Test). Although immunofluorescence has not been calibrated to underlying MBP concentration, the observation that GABA axons have more MBP than neighboring non-GABA axons was statistically significant and qualitatively consistent across multiple experiments.

## The cytoskeletal composition of myelinated GABA axons is different from myelinated non-GABA axons and from unmyelinated GABA axons

The most striking difference between GABA and non-GABA myelinated axons is in their cytoskeleton. Immunostaining for the neurofilament heavy chain and for alpha tubulin revealed that myelinated GABA axons are rich in neurofilaments and have relatively low microtubule content, while the cytoskeleton of myelinated non-GABA axons is dominated by microtubules (*Figure 9*). In addition, a larger fraction of the tubulin in myelinated GABA axons appears to be acetylated. While GABA myelinated axons contain on average only 66% of the alpha tubulin found in non-GABA myelinated axons, they have 86% of the acetylated alpha tubulin content of non-GABA myelinated axons. These differences in the cytoskeletal composition of myelinated axons were observed throughout the cortical layers.

The differences in cytoskeletal composition of myelinated axons could be due to a general difference between GABA and non-GABA axons, regardless of their myelination status. To explore this possibility, we identified individual axons that contained both a myelinated and a non-myelinated portion within the dataset volume. Nodes of Ranvier were excluded and only non-myelinated stretches longer than 4 μm were considered. This analysis revealed several interesting differences (*Figure 9D*). First, the cytoskeletal composition of axons is indeed related to the presence of GABA. Both the myelinated and the unmyelinated regions of the GABA axons have higher neurofilament content and lower microtubule content compared to non-GABA axons. There are also significant differences in cytoskeletal content between the different regions of GABA axons, where myelinated

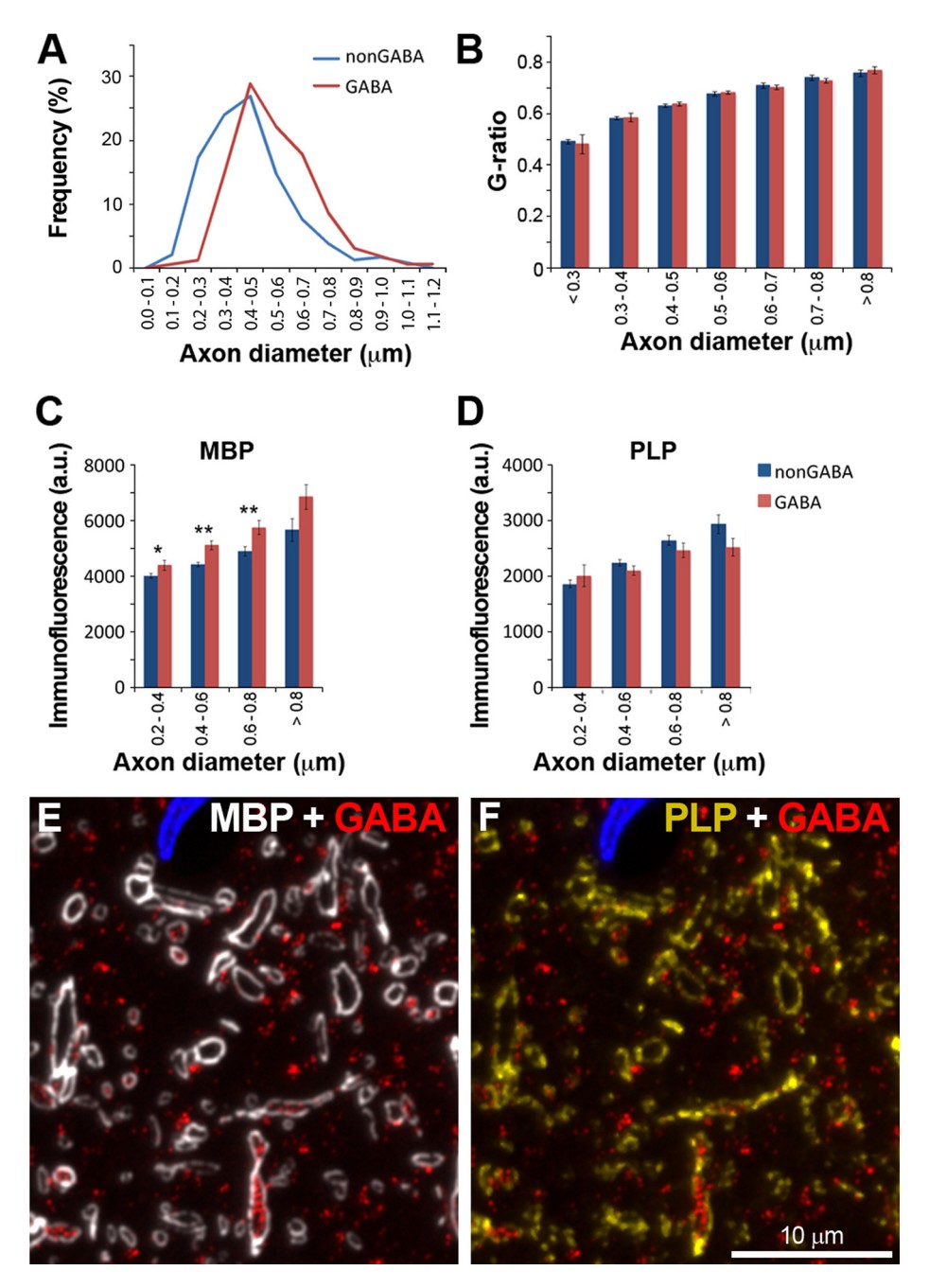

**Figure 8.** Similarities and differences between the myelin of GABA and non-GABA axons. (**A**) GABA axons have thicker axons on average. Frequency distribution plot of the thickness measurements of 238 non-GABA and 163 GABA axons. (**B**) GABA and non-GABA axons have similar g-ratios (mean ± standard error, n=238 non-GABA and 163 GABA axons, Mann-Whitney U Test). (**C**) The myelin of GABA axons contains significantly more myelin basic protein (MBP) than non-GABA axons (mean ± standard error, n=489 non-GABA and 254 GABA axons). (**D**) There are no significant differences in the PLP content of myelinated axons. All the analyses for this Figure were performed in cortical layers 4 and 5, which have high density of GABA myelinated axons. Asterisks indicate statistically significant differences (**p<0.01, *p<0.05, Mann-Whitney U Test). (**E**) and (**F**) show an example of MBP and PLP immunofluorescence on a single section from the same dataset as analyzed in **C** and **D**.

The following source data is available for figure 8:

**Source data 1.** Data values and statistics underlying *Figure 8*.

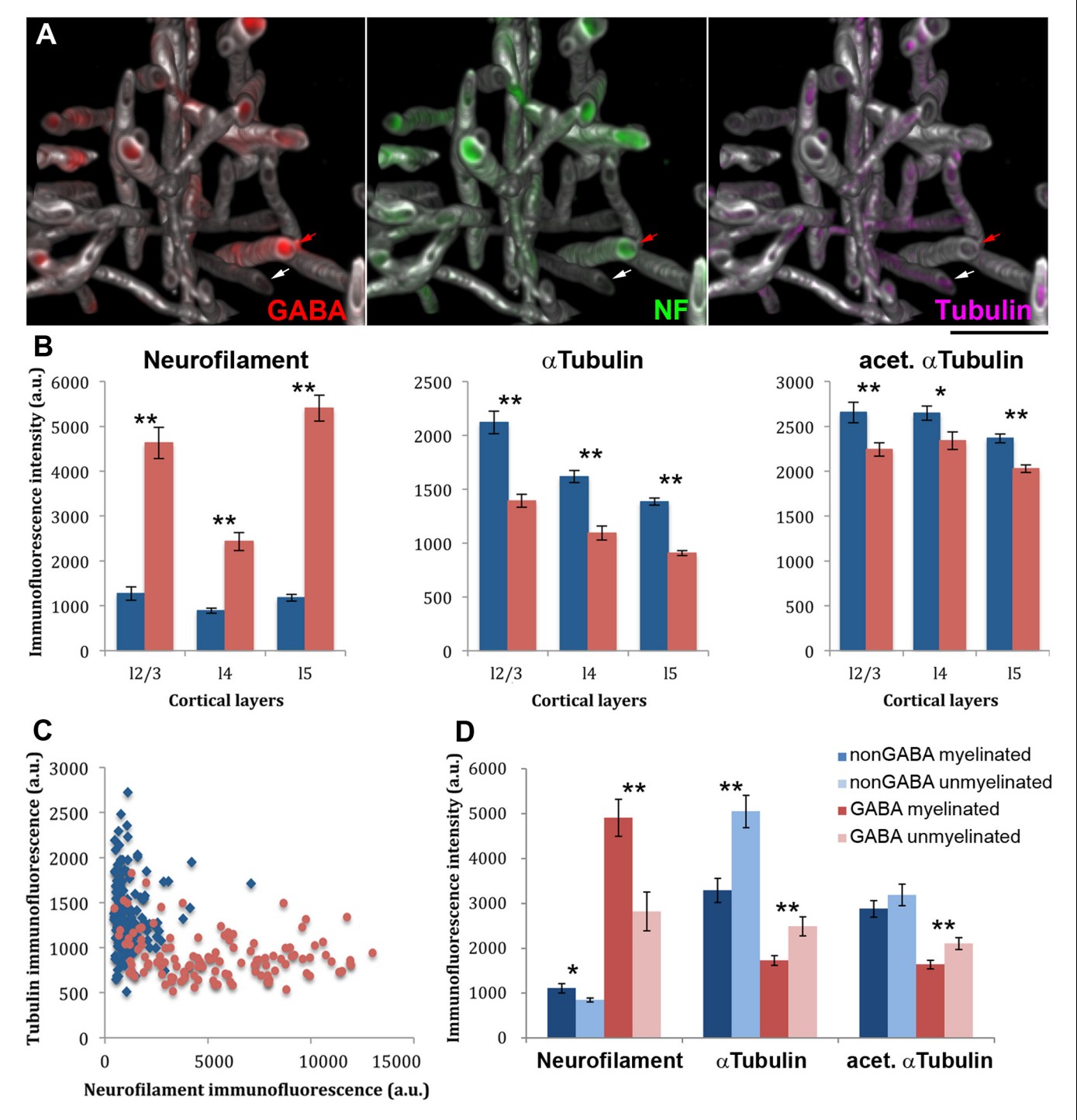

**Figure 9.** The cytoskeletal composition of myelinated GABA axons is different from myelinated non-GABA axons, as well as from unmyelinated GABA axons. (**A**) The reconstructed small volume (16 × 16 × 4.5 μm) from layer 5 of the mouse barrel cortex contains GABA axons (red), which are enriched in neurofilaments (NF-H, green), and non-GABA axons where microtubules (αtubulin, magenta) predominate. For clarity, only immunofluorescence signal within myelinated axons (MBP, white) is displayed. Red arrow points to a GABA positive axon and the white arrow to a GABA negative axon. (**B**) The immunofluorescence intensity (mean ± standard error) for neurofilament heavy chain, αtubulin and acetylated αtubulin within myelin profiles is compared between GABA (red) and non-GABA axons (blue). A larger volume spanning layers 2/3, 4 and 5, and including the region presented in A was analyzed. The differences are statistically significant (p<0.01, Mann-Whitney U Test) in all cortical layers analyzed (layers 2/3: 80 nonGABA and 68 GABA axons; layer 4: 121 nonGABA and 65 GABA axons; and layer 5: 146 non-GABA and 119 GABA axons). (**C**) Analysis of the cytoskeletal content of the two types of axons from layer 5 (146 non-GABA and 119 GABA axons). (**D**) Comparison of myelinated vs. unmyelinated stretches of axons. Individual axons

*Figure 9 continued on next page*

*Figure 9 continued*

which contained both a myelinated and a non-myelinated portion within the dataset volume were analysed (11 non-GABA and 20 GABA axons from layers 2/3, 4 and 5). Nodes of Ranvier were excluded from the analysis. Statistical differences using paired t-test are reported: *p<0.05, **p<0.01).

The following source data is available for figure 9:

**Source data 1.** Data values and statistics underlying *Figure 9*.

portions have higher neurofilament and lower microtubule content compared to the unmyelinated portions of the same axons. While excitatory axons show similar trends of differences in cytoskeletal composition depending on myelination, the differences are more pronounced for GABA axons, especially regarding neurofilament content.

## Discussion

The present work is the first to survey the targets of myelin in neocortex using recently developed high resolution, large-volume imaging methods (*Briggman and Bock, 2012*; *Kleinfeld et al., 2011*; *Economo et al., 2016*; *Micheva and Smith, 2007*). These methods reveal that a surprisingly large fraction of myelin in cortical gray matter ensheathes the axons of a single inhibitory neuron subtype, the parvalbumin-positive basket cell. Based on cellular morphology and postsynaptic targets, other PV-expressing neuron types were essentially ruled out, and we found minimal evidence for the myelination of non-PV inhibitory axons. Although this cell-type-specificity of axon myelination was unexpected, especially since myelin has generally been thought to ensheath long-range axons of excitatory neurons, it is nevertheless consistent with earlier observations of individual myelinated inhibitory axons, including basket cell axons (*Peters and Proskauer, 1980*; *DeFelipe et al., 1986*; *Somogyi et al., 1983*; *Keller and White, 1987*).

The more comprehensive survey presented in the present work was achieved through analysis of data arising from two different imaging modalities: array tomography (*Micheva and Smith, 2007*; *Collman et al., 2015*) and large-scale serial-section electron microscopy (*Bock et al., 2011*). Work based on each method was initiated independently and in parallel, and initial annotation and analysis was performed blind to preliminary results arising from the other method. The results from AT and volume EM were complementary. The AT data spanned all cortical laminae and revealed that many myelinated axons in mouse somatosensory cortex (almost half of layer 2/3, and a quarter of layer 4 myelinated axons) contain the inhibitory neurotransmitter GABA. Practically all of these GABA axons also contain PV. AT-based comparison of GABA and nonGABA myelinated axons revealed several important differences. The nodes of Ranvier and the internodes are significantly shorter on GABA axons. In addition, AT multiplex imaging detected molecular differences, such as an enrichment of MBP in the myelin of GABA axons, as well as a high neurofilament and low microtubule content of their cytoskeleton.

The volume EM data provided independent evidence that half of the categorized myelinated axons in layer 2/3 of mouse visual cortex are inhibitory, as indicated by the symmetric pre- and post-synaptic densities of the synapses that they formed after exiting the myelin sheath (*Colonnier, 1968*; *Peters et al., 1991*; *Gray, 1959*). Immunolabeling of myelin in a PV-Cre reporter line showed that about half of the myelin in layer 2/3 ensheathes PV+ axons, and similar experiments in VIP- and SOM-Cre lines (which comprise almost all non-PV interneurons [*Xu et al., 2010*]) failed to show any myelinated axons arising from VIP+ neurons and very few (if any) from SOM+ neurons. Ultrastructural analysis also allowed us to deduce which of the two PV-expressing subtypes in neocortex was myelinated, by revealing that the myelinated axons' postsynaptic targets were largely non-axonal, instead targeting the dendrites and somata of pyramidal cells. This finding ruled out PV-positive chandelier cells (which synapse onto axons), and strongly suggested that the other predominant PV subtype, basket cells, is the myelinated subtype. PV basket cells comprise about half of all neocortical interneurons (*Markram et al., 2004*). Taken together, the quantitative and qualitative consistency between the AT, volume EM, and immunohistochemistry datasets provides strong evidence that PV + basket cell axons are the predominant myelinated inhibitory axons in neocortex.

Our axon reconstructions showed that the distribution of myelin on basket cell axons is patchy and seemingly localized to the axon arbor near the cell body (e.g. *Figures 3D–F*, *5A–C*). This patchiness qualitatively resembles the distribution of myelin on the descending axons of excitatory pyramidal cells measured in the same EM data set (*Tomassy et al., 2014*), and is consistent with an observation of patchy myelination of an inhibitory neuron axon in an earlier study in mouse somatosensory cortex (*Keller and White, 1987*). The concentration of myelin on the proximal arbor is also consistent with more complete reconstructions of individual filled cells in cat (*Somogyi et al., 1983*) and macaque (*DeFelipe et al., 1986*).

There are several caveats and limitations to this study. First, our experiments allow for the possibility that a small fraction of somatostatin or other non-PV axons are myelinated. We found by AT that ~2% of myelinated GABA axons are PV-negative, and examination of the transgenic mouse line SOM-Cre;Ai9-TdTom revealed a small percentage of TdTomato-positive myelinated axons. However, the lack of PV immunostaining in some myelinated axons may reflect limitations of immunodetection rather than proving that these axons are PV-negative. Furthermore, the SOM-Cre-line used in this study was found to have a small but consistent false-positive expression pattern, where Cre is expressed in neurons that immunohistochemically are somatostatin-negative but PV-positive (*Hu et al., 2013*). Although additional studies of somatostatin-positive interneurons, as well as other inhibitory interneuron subtypes, will be needed to resolve this issue, we believe it does not materially affect our finding that the great preponderance of myelinated inhibitory axon profiles in layer 2/3 arise from parvalbumin positive basket cells.

A second limitation stems from the incomplete characterization of the distribution of myelin on the full arbors of inhibitory neuron axons. Myelinated axon profiles were traced until they exited the myelin sheath and formed a sufficient number of synapses to categorize the axon as excitatory or inhibitory, at which point tracing was halted (except in two cases, *Figure 5A and B*, where the entire arbor within the 450 × 350 × 50 µm EM volume was traced). The result was a set of axon fragments (*Figure 5C*) that cannot be positively identified according to morphological type. Their identification as arising from basket cells is based on a combination of immunolabeling results (that myelinated inhibitory axons are nearly always PV+) with the identification of the fragments' post-synaptic targets as being almost entirely non-axonal (so the axons do not arise from chandelier cells, the other PV+ cell type besides basket cells). Ideally, direct confirmation of this deduction would also be obtained from complete volume EM axonal arbor reconstructions and subsequent confirmation of morphological type. However, the additional tracing effort to achieve this would be substantial (*Helmstaedter, 2013*), and, perhaps more importantly, such an investment might better be made in a larger cortical EM volume (e.g [*Lee et al., 2016*]), so that a greater fraction of each neuron's axonal arbor could be reconstructed. Such an approach would also permit assessment of whether all PV-containing basket cells have myelinated axons. Considering the functional and morphological heterogeneity of PV basket cells (*Kawaguchi and Kubota, 1997*; *Wang et al., 2002*; *Li and Huntsman, 2014*), it is possible that some subtypes preferentially form myelinated axons.

Consistent with the possibility that not all basket cells (or basket cell subtypes) are myelinated, the fraction of somatic post-synaptic targets made by the inhibitory myelinated axons described here is at the low end of the previously reported range for cortical basket cells. The majority of myelinated inhibitory axons in our study contact dendritic shafts (53%) and spines (32%), and only 13% of the targets are cell bodies. These proportions are similar to those found in a study in rat frontal cortex of four individually filled fast-spiking basket cells, which made on average 53% of synapses onto dendritic shafts, 29% onto spines, and 18% onto cell bodies (*Kawaguchi and Kubota, 1998*). However, the four cells showed wide variability in the proportion of their somatic targets, which ranged between 4% and 35%. Studies in cat neocortex also reveal a wide variability between individual basket cells (between 20% and 69% of synapses targeting somata) (*Somogyi et al., 1983*; *Kisvarday et al., 1987*; *Tamás et al., 1997*). These findings in neocortex contrast with measurements in hippocampus, where more than 50% of the synaptic targets of PV basket cells are cell bodies, and dendritic spines are only occasionally targeted (*Seress and Ribak, 1990*; *Gulyás et al., 1993*; *Halasy et al., 1996*). The fraction of inhibitory post-synaptic targets made by the axons in this study (4.5%) is similar to, but somewhat lower than that previously observed in rat (8%) and cat (9%) neocortex (*Somogyi et al., 1983*; *Kawaguchi and Kubota, 1998*). This variability is consistent with a recent study in mouse visual cortex which found wide variation between individual animals in the fraction of synapses made onto inhibitory interneurons by layer 2/3 pyramidal cells (*Bopp et al.,*

*2014*). Overall, the anatomical literature briefly described above shows that the post-synaptic targets of a given cell type can vary widely between species, brain regions, animals, and individual neurons. Determining whether discrete basket cell subtypes underlay any of this variability will require further study.

Our findings raise the question: what is the role of myelin on inhibitory interneurons in neocortex? The patchy distribution of myelin along GABA axons and the lack of correlation between myelin thickness and axon diameter imply that increased conduction velocity may not be the main consequence of this myelination. Rather, the presence of myelin may be due to the unique cellular properties and circuit functions (*Hu et al., 2014*) of PV basket cells. Nearly all cortical PV basket cells are fast-spiking (*Kawaguchi and Kubota, 1997*; *Galarreta and Hestrin, 1999*, *2002*; *Gibson et al., 1999*), and fast-spiking neurons have very high tonic activity compared to other cell types in cortex (*Gentet et al., 2010*; *Hofer et al., 2011*) — they generate long trains of action potentials with very high frequency (>100 Hz) (*Kawaguchi and Kubota, 1997*; *Galarreta and Hestrin, 1999*, *2002*; *Gibson et al., 1999*). Fast-spiking cells exhibit a number of biophysical specializations that enable high firing rates but incur an increased energy cost per action potential (*Hu and Jonas, 2014*; *Carter and Bean, 2009*; *Hasenstaub et al., 2010*). Axonal myelination could help with these high energy demands in two ways: first, myelin is known to improve the energy efficiency of action potential propagation (*Hartline and Colman, 2007*). Secondly, myelin can provide metabolic support to the axon by supplying lactate as an energy source (*Rinholm et al., 2011*; *Lee et al., 2012*; *Fünfschilling et al., 2012*; *Morrison et al., 2013*). Thus, myelin could reduce the basket cells' metabolic needs and provide an additional external energy supply from oligodendrocytes.

At the circuit level, PV basket cells participate in regulating excitatory/inhibitory balance, in neuronal synchronization and cortical rhythm generation, and in plasticity associated with experience and learning (*Hu et al., 2014*). Could basket-cell myelin hasten the delivery of synaptic input to their postsynaptic targets to an extent that would be functionally relevant? No estimates of conduction velocities of cortical myelinated PV axons exist in the literature; however, in young (~P30) mice, PV axon conduction velocities of ~0.5 m/s have been recorded (*Casale et al., 2015*; *Li et al., 2014*). The majority of PV axons at this age are most likely unmyelinated, since cortical myelination is not complete until much later (after P60) (*Barrera et al., 2013*; *Mengler et al., 2014*). Using this conduction velocity estimate, an unmyelinated PV axon would conduct a spike in 0.4 ms to synapses located 200 μm away, which was the typical path length from cell body to proximal synapses in our EM-based reconstructions. Thus, any reduction in the spike arrival time to proximal synapses due to myelination will inevitably be in the sub-millisecond range, likely too small a difference to affect post-synaptic dendritic integration or spike-timing-dependent plasticity, given the much slower timescales of those functions (*Dan and Poo, 2006*; *Feldman, 2012*; *Stuart and Spruston, 2015*). However, spike time arrival differences on a sub-millisecond timescale might affect the magnitude of coordinated oscillations (*Pajevic et al., 2014*), which are a known circuit role of basket cells.

The present study also bears on the genesis and dynamics of myelination. Cortical myelin is dynamically remodeled during learning and development (*McKenzie et al., 2014*; *Bergles and Richardson, 2016*; *Makinodan et al., 2012*) and it redistributes during normal aging, where internode length decreases with age in primates (*Peters, 2009*). It is possible that these dynamics may differ between the myelin wrapping inhibitory and excitatory axons. This could be due to differences arising from local interactions between oligodendrocytic processes and individual axons, or from a possible allocation of distinct oligodendrocytes (or oligodendrocyte subtypes [*Tomassy et al., 2016*]) to axons of inhibitory or excitatory neurons (*de Hoz and Simons, 2015*).

Further studies will also be needed to address the interesting question of myelin diversity at the molecular level. Between the two major proteins of myelin, our experiments reveal that inhibitory axons have higher MBP content than excitatory axons, while the proteolipid protein (PLP) content appears similar. Because MBP is one of the potential target antigens in MS (*Wucherpfennig and Strominger, 1995*; *Fujinami and Oldstone, 1985*; *Garg and Smith, 2015*), this finding has important implications for pathologies of myelin. It is likely that other differences in the protein or lipid content of myelin exist, considering the accumulating evidence for CNS oligodendrocyte heterogeneity (*Tomassy et al., 2016*). Characterizing these differences as a function of neuronal type will be an important aspect of future studies.

The abundance of myelinated inhibitory axons in cortical gray matter also raises the possibility of their involvement in myelin perturbations in normal aging (*Peters, 2009*) and disease (*Fields, 2008*;

*Popescu and Lucchinetti, 2012*; *Calabrese et al., 2015*; *Wingerchuk et al., 2015*; *Gordon et al., 2014*; *Lucchinetti et al., 2011*; *Kang et al., 2013*). For example, gray matter myelin degeneration is observed in the early stages of Alzheimer's disease (AD) (*Bartzokis, 2004*; *de la Monte, 1989*), and, in relapsing remitting multiple sclerosis (RRMS), the presence of cortical lesions is strongly correlated with seizure occurrence (*Calabrese et al., 2008*). Several other diseases predicated on CNS demyelination, including neuromyelitis optica, acute disseminated encephalomyelitis, and progressive multifocal leukoencephalopathy may also display seizure activity (*Jarius et al., 2014*; *Hynson et al., 2001*; *Lima et al., 2006*). Demyelination of PV interneurons may perturb the balance of large-network neuronal excitation and inhibition, biasing neocortical microcircuits toward seizure.

By combining data from two independent efforts, we show here that a large fraction of cortical myelin is allocated to the axons of inhibitory neurons, specifically to PV basket cells. This work alters our understanding of the distribution of myelin in neocortical gray matter, and opens new avenues in basic neurobiology, systems neuroscience, and neurological disease research. Finally, this study leverages an openly available EM data set (*Burns et al., 2013*; *Martone et al., 2002*) that has been used in several other studies of neocortex (*Tomassy et al., 2014*; *Bopp et al., 2014*) since its publication (*Bock et al., 2011*), illustrating the value of image data sharing in biomedical research.

## Materials and methods

### Array tomography

#### Lowicryl freeze-substitution tissue preparation

All animal procedures were performed according to NIH and University of North Carolina guidelines. After deep anesthesia with pentobarbital, adult mice (3 to 4 months old) were perfusion-fixed with a mixture of 2% glutaraldehyde/2% formaldehyde, dissolved in 0.1 M phosphate buffer (pH 6.8). Brains were removed and postfixed overnight at 4°C in the same fixative. 200 μm-thick Vibratome sections were collected, incubated on ice on a shaker with 0.1% $CaCl_2$ in 0.1 M sodium acetate for 1 hr, then cryoprotected through 10% and 20% glycerol, and overnight in 30% glycerol in sodium acetate solution. The next day, small tissue chunks from S1 neocortex were dissected out and quick-frozen in a dry ice/ethanol bath. Freeze-substitution was performed using a Leica AFS instrument with several rinses in cold methanol followed by substitution in a 2–4% solution of uranyl acetate in methanol, all at -90°C. After 30 hr incubation, the solution was slowly warmed to −45°C and infiltrated with Lowicryl HM-20 over 2 days. Capsules containing tissue chunks were then exposed to UV during gradual warming to 0°C. Polymerized capsules were removed from the AFS apparatus and further exposed to UV at room temperature for an additional day, to complete curing of the plastic.

Ribbons were prepared and imaged using standard methods of array tomography (*Micheva et al., 2010*). 70 nm-thick serial sections of the embedded plastic block were cut on an ultramicrotome (Leica Ultracut EM UC6, Leica Microsystems, Wetzlar, Germany) and mounted on gelatin-coated or carbon-coated coverslips. Gelatin-subbed coverslips were used for most of the immunofluorescent array tomography. Carbon-coated coverslips, which provide better adhesion of the sections and are conductive, were used for immunofluorescent experiments with more than 3 staining and imaging cycles and for conjugate immunofluorescent / SEM experiments.

#### Immunofluorescent array tomography

Sections were processed for standard indirect immunofluorescence and imaged on an automated epi-fluorescent microscope (Zeiss AxioImager Z1, Zeiss, Oberkochen, Germany) using a 63x Plan-Apochromat 1.4 NA oil objective, as described in *Micheva et al. (2010)*. Antibodies were obtained from commercial sources and their use has been reported in numerous studies (*Tomer et al., 2014*; *Beirowski et al., 2014*; *Curchoe et al., 2010*; *Hodgson et al., 1985*; *Puthussery et al., 2011*; *Marek et al., 2010*; *González-Albo et al., 2001*; *Álvarez-Quilón et al., 2014*; *Kamura et al., 2011*). We performed AT specific controls for all antibodies. (*Supplementary file 1* and *Figure 1—figure supplement 1*). In addition, MBP immunofluorescence was compared to myelin from SEM images of the same sections (*Figure 1—figure supplement 2*). The sources and dilutions of all primary antibodies used in this study can be found in *Table 2*. The primary antibodies were applied for 2 hr at room temperature, or overnight at 4°C. Alexa dye conjugated secondary antibodies, highly cross-adsorbed (Life Technologies, Carlsbad, CA), were used at a dilution of 1:150 for 30 min at room

**Table 2.** Primary antibodies used in this study.

| Antigen | Host | Antibody source | Dilution | RRID |
|---|---|---|---|---|
| MBP | Chicken | AVES MBP | 1:200 | RRID:AB_2313550 |
| GABA | Guinea pig | Millipore AB175 | 1:5000 | RRID:AB_91011 |
| Parvalbumin | Rabbit | SWANT PV28 | 1:300 | RRID:AB_2315235 |
| NF-H | Chicken | AVES NFH | 1:100 | RRID:AB_2313552 |
| NF-L | Chicken | AVES NFL | 1:100 | RRID:AB_2313553 |
| αTubulin | Rabbit | Abcam ab18251 | 1:100 | RRID:AB_2210057 |
| Acetylated αtubulin | Mouse | Sigma T6793 | 1:100 | RRID:AB_477585 |
| Glutamine Synthetase | Mouse | BD Biosciences 610517 | 1:25 | RRID:AB_397879 |
| PLP | Chicken | AVES PLP | 1:100 | RRID:AB_2313560 |

temperature. One difference from the published protocols was the application of sodium borohydride (1% in Tris buffer for 3 min) as the first step of the immunofluorescent labeling to reduce nonspecific staining and autofluorescence. For some samples, after the sections were imaged, the antibodies were eluted using a solution of 0.2 M NaOH and 0.02% SDS and new antibodies were reapplied. Several rounds of elution and restaining were applied to create a high-dimensional immunofluorescent image.

To define the position list for the automated imaging, a custom Python-based graphical user interface, MosaicPlanner (available at http://smithlabsoftware.googlecode.com), was used to automatically find corresponding locations across the serial sections. Images from different imaging sessions were registered using a DAPI stain present in the mounting medium. The images from the serial sections were also aligned using the DAPI signal. Both image registration and alignment were performed with the MultiStackReg plugin in Fiji (*Schindelin et al., 2012*).

## Scanning electron microscopy

After IF imaging, samples were rinsed with water and poststained with 5% aqueous uranyl acetate (UA) for 30 min and freshly prepared and filtered 1% *Reynolds' lead citrate* for 1 min. The coverslips were attached to 50 mm pin mounts (Ted Pella, Redding, CA) using carbon paint. Ribbons were imaged on a Zeiss Sigma field emission scanning electron (FESEM) microscope using the backscatter detector at 5–8 KeV. The corresponding regions of the sample were located using the correlations between the DAPI stain from the immunofluorescence and the ultrastructure of the nuclei as seen in the SEM.

## Registration of light microscopy and scanning electron microscopy

Light and electron microscopy images were registered with the TrakEM2 plugin (*Cardona et al., 2012*) within Fiji. To identify the same structures in images from both acquisition systems, DAPI fluorescence images were histogram-normalized to make the spatial structure in both the dim autofluorescence and brighter DAPI fluorescence equally apparent. This is useful because variations in the dim autofluorescence correspond to ultrastructural features visible in the electron microscope, such as large dendrites, mitochondria, and myelin. Bright DAPI fluorescence corresponds to the ultrastructurally identified heterochromatin in cell nuclei. Several corresponding features (4–6) in the DAPI images and the EM images were used to fit a similarity transformation (rigid rotation plus uniform scaling). This transformation was automatically applied to the other light microscopy images. Our image reconstruction tools are all available at smithlabsoftware.googlecode.com.

## Immunofluorescent image analysis and statistics

Volumes from the somatosensory cortex of 3 mice were used for analysis. Most volumes comprised of approximately 60 serial sections (range of 43 to 81 sections) and included all cortical layers. For each layer, a field of view of approximately 135 by 130 μm was analyzed. Immunofluorescence measurements were performed on raw images using FIJI. MBP immunofluorescence was used to define

regions of interest (ROI), which were either the myelin sheath for measurements of MBP and PLP signal, or the axon under the myelin sheath for measurement of axonal immunofluorescence for GABA, PV, and cytoskeletal proteins. The mean gray value of immunolabels was compared between GABA and nonGABA axons from the same coverslip, using the non-parametric Mann-Whitney U Test. Axons were classified as GABA positive or PV positive based on an empirically determined threshold for each experiment, as shown in *Figure 3—figure supplement 1*. The distribution of GABA immunofluorescence showed a peak of low immunofluorescence corresponding to background, followed by a clearly defined second broad peak corresponding to GABA immunopositive axons. PV immunofluorescence exhibited a less clear separation between the background and immunofluorescence peaks, however, as expected the great majority of GABA immunonegative axons were also classified as immunonegative for PV. With the thresholds set in the example of *Figure 3—figure supplement 1*, only 3.1% of GABA negative axons were classified as PV positive, and 1.7% of GABA positive axons were classified as PV negative. AT analysis and statistics were done in the Smith laboratory prior to knowledge of the volume EM or immunohistochemistry results from the Bock laboratory.

## Tracing of volume electron microscopy data

A publicly available (*Burns et al., 2013*; *Martone et al., 2002*) 450 × 350 × 50 µm EM dataset from the upper layers of visual cortex (*Bock et al., 2011*) in an adult Thy1-YFP-H (*Feng et al., 2000*) mouse (9–14 months of age) was used. The images were obtained by transmission electron microscopy of serial thin sections, resulting in a ~10 TB dataset comprised of ~4 × 4 × 45 nm voxels. To sample a subset of myelinated axons in the volume systematically, a section near the center of the image volume was selected, and all myelinated axon profiles were annotated in TrakEM2 with 'seed' points for further tracing (*Figure 2*). Myelinated profiles near the edge of the imaged area were not seeded, to reduce the probability of the axon exiting the volume without making synapses. To categorize each myelinated axon as arising from an inhibitory or excitatory neuron, each seed was manually traced into the EM volume until the axon lost its myelin sheath and formed at least two synapses. The synapses were categorized as inhibitory or excitatory depending on whether the pre- and postsynaptic densities were symmetric (equal in thickness) or asymmetric (with a thicker postsynaptic density), respectively. Occasionally, individual synapses could not be definitively characterized as symmetric or asymmetric. This occurred most often when the section plane was oblique or parallel to the plane of the synapse. In these cases, tracing was continued until sufficient additional synapses were reached to make a definitive determination of axon type. This work was performed prior to the tracers' knowledge that the array tomography work existed. Categorization of axons as inhibitory or excitatory based on synapse ultrastructure was done in the Bock laboratory prior to knowledge of the AT results from the Smith laboratory.

All tracing of fine unmyelinated axon branches and synaptic classifications were reviewed by a highly experienced tracer (DDB) for correctness. In all cases where a traced axon reached a cell body (*Figure 2*, *Figure 2—figure supplements 1*, *2*), the categorization of the axon as excitatory or inhibitory based on synaptic ultrastructure was found to agree with somatic structure and synaptic interactions: excitatory axons were only found to arise from pyramidal neurons receiving only inhibitory synapses on the cell body, and inhibitory axons were only found to arise from non-pyramidal neurons with both inhibitory and excitatory synapses on the cell body (*Peters et al., 1991*). Axons that could not be classified generally failed to unmyelinate within the bounds of the EM-imaged volume and had an orientation normal to the EM section plane (*Figure 2—figure supplement 3*). On this axis, the EM image volume is only ~50 µm thick, reducing the probability that these axons could be traced to an unmyelinated, synapse-forming portion of the axonal arbor.

The postsynaptic target at each annotated synapse was also classified as excitatory or inhibitory using ultrastructural criteria. The dendritic shafts of inhibitory neurons have much lower spine densities and receive many more asymmetric synapses than those of excitatory neurons (*Bock et al., 2011*; *McGuire et al., 1991*) (*Figure 6E*), allowing for unambiguous categorization. When the postsynaptic target was a dendritic shaft, categorization was therefore straightforward. When the target was a spine, the spine was traced through its neck to the shaft, and categorization was then based on shaft ultrastructure. In rare cases, the spine could not be traced to the shaft and in this case no attempt was made to categorize the postsynaptic target as excitatory or inhibitory (*Table 1*, 'Uncategorized dendritic spines'). Postsynaptic axons were identified based on a combination of the following criteria: bundled microtubules and the presence of a characteristic dense layer beneath the

plasma membrane (*Peters et al., 1991*); their presynaptic relationships to other neurons (*Figure 5F*); and their eventual entrance into myelin sheaths. Postsynaptic targets were identified after pooling of data between the two laboratories had begun.

Two myelin seeds were arbitrarily selected for complete tracing (*Figure 5A,B*). Rather than halting tracing of these axons once they could be unambiguously categorized as inhibitory or excitatory, their arbors were traced to completion within the limits of the EM-imaged volume. In one case, the inhibitory cell body was reached (*Figure 5A*). All synapses were annotated and their postsynaptic targets were categorized as with the partially traced axons. The distribution of postsynaptic targets was nearly identical between the completely and partially traced axons (*Table 1*).

## Immunohistochemistry of transgenic mice

### Tissue preparation

All animal procedures were performed according to HHMI-JFRC animal care and use protocols. Following isoflurane anesthesia, adult transgenic mice (male *Pvalb-ires-Cre;Ai9, Sst-ires-Cre;Ai9*, and *Vip-ires-Cre;Ai9* [*Taniguchi et al., 2011*; *Madisen et al., 2010*]) between 1.5 and 4 months of age were fixed by cardioperfusion with a solution of fresh mixed 4% paraformaldehyde (J.T. Baker, Capitol Scientific) in 1x phosphate buffered saline (pH 7.4). Serial 40 μm thick Vibratome sections through S1 were collected, which were subsequently post-fixed in the same fixative solution overnight at 4°C. Slices were then triply washed in 1x PBS before blocking and permeabilization with 10% normal goat serum and 0.5% Triton X-100 in 1x PBS (Sigma-Aldrich, St. Louis, MO) for 12 hr at room temperature on a benchtop shaker. A 1:400 solution of rat anti-myelin basic protein antibodies (aMBP; Millipore MAB386) was then applied for 24 hr at room temperature, followed by a triple wash with 1x PBS and incubation in a 1:400 solution of goat anti-rat Alexa 488 conjugated antibodies (Invitrogen A11006) overnight at 4°C. Slices were then triple washed once again before overnight counterstaining with DAPI (Invitrogen, Waltham, MA) and subsequent mounting under #1.5 coverglass (Zeiss) with glycerol-based Vectashield mounting medium (Vector Labs, Burlingame, CA).

### Confocal microscopy

Confocal images of prepared samples were acquired on a Zeiss LSM 710 using a 40x Plan-Apochromat 1.4 N.A. oil immersion objective. Three-color imaging was performed by sequential acquisitions with 405 nm, 488 nm, and 561 nm lasers and pinhole parameters set to 1 Airy unit for any given laser. Detection was tuned to each fluorophore's specific emission frequency using a 32-channel detector without the use of filters. Laser powers were scaled with increasing acquisition depth by spline interpolation to maintain constant fluorophore visibility. Images were acquired through a large region of S1 from layer 1 through layer 6 in sequential overlapping tile mosaics using the Multitime32 macro (Zeiss) and subsequently stitched using the Grid/Collection stitching plugin for Fiji (*Preibisch et al., 2009*).

### Quantification of fluorescence labeled myelin and transgene reporter axons

Three image stacks, each from a separate animal, were collected for the *PV-ires-Cre;Ai9* mouse line, while three image stacks were acquired from a single mouse from each of the *SOM-ires-Cre;Ai9* and *VIP-ires-Cre;Ai9 lines*. In the PV and VIP image stacks, an 85 × 85 μm ROI in layer 2/3 of S1 was selected for annotation and quantification. In the SOM image stack, a 200 × 200 μm ROI in layer 2/3 of S1 was quantified, to better quantify the frequency of rarely myelinated SOM-Cre+ axonal profiles. The z-plane with the most robust myelin labeling, typically within 15 μm of the surface of the vibratome slice, was selected for annotation of myelinated axons and *Ai9* reporter labeled neurites. Annotation was performed using the TrakEM2 plugin (*Cardona et al., 2012*) for Fiji and consisted of manual treeline skeletonization of fluorescent neurites over multiple z-planes adjacent to the target ROI plane. The total number of myelinated axons colabeled for PV (as represented by Ai9-fluorophore expression) was quantified. Colocalization of myelin and the reporter fluorophore along a visible neurite spanning multiple z-planes was considered as a myelinated reporter-labeled axon. The fraction of myelinated reporter labeled axons was tallied for each ROI, and pooled within each reporter line. VIP fractions were not annotated and quantified since by careful visual inspection, no reporter-labeled neurites were found to colocalize with myelin signal. For preparation of figures, Adobe Photoshop was used to smooth and adjust gamma of raw confocal data, and to false-color

myelinated profiles that were positive for either PV (*Figure 4C–D*) or SOM (*Figure 4I–J*). Immunohistochemistry was done in the Bock laboratory prior to knowledge of the AT results from the Smith laboratory.

## Acknowledgements

The authors are grateful to Dr. Richard Weinberg for providing the material used in AT experiments. We thank Maita Esteban and Dr. Jong-Cheol Rah for EM tracing and help with immunohistochemistry analysis, and Nelson Spruston, Jeff Magee and Erik Bloss for helpful comments on the manuscript. This work was supported by NIH grants (R21MH099797, R01NS75252, R01NS092474 to SJS) and by the Howard Hughes Medical Institute (DDB).

## Additional information

### Competing interests

KDM, SJS: Has founder's equity interests in Aratome, LLC (Menlo Park, CA), an enterprise that produces array tomography materials and services. Also listed as inventor on two US patents regarding array tomography methods that have been issued to Stanford University (US patents 7,767,414 and 9,008,378). The other authors declare that no competing interests exist.

### Funding

| Funder | Grant reference number | Author |
| --- | --- | --- |
| National Institutes of Health | R21MH099797 | Stephen J Smith |
| National Institutes of Health | R01NS75252 | Stephen J Smith |
| National Institutes of Health | R01NS092474 | Stephen J Smith |
| Howard Hughes Medical Institute | | Davi D Bock |

The funders had no role in study design, data collection and interpretation, or the decision to submit the work for publication.

### Author contributions

KDM, Conceived and designed the study, Performed IF in AT experiments, Acquisition of data, Analysis and interpretation of data, Drafting or revising the article; DW, Performed immunohistochemistry experiments, Acquisition of data, Analysis and interpretation of data, Drafting or revising the article; BDM, Contributed to data analysis and interpretation, Analysis and interpretation of data, Drafting or revising the article; EP, Traced volume EM data, Analyzed immunohistochemistry data, Acquisition of data, Analysis and interpretation of data, Drafting or revising the article; JAB, Performed SEM in AT experiments, Acquisition of data, Drafting or revising the article; SJS, Contributed to data analysis and interpretation, Drafting or revising the article; DDB, Conceived and designed the study, Acquisition of data, Analysis and interpretation of data, Drafting or revising the article

### Author ORCIDs

Dylan Wolman, http://orcid.org/0000-0002-5012-1690
Davi D Bock, http://orcid.org/0000-0002-8218-7926

### Ethics

Animal experimentation: The tissue for the array tomography experiments was provided by Dr. Richard Weinberg, University of North Carolina (UNC). All animal procedures were performed according to NIH and UNC guidelines with a protocol (#13-258.0) approved by the UNC Institutional Animal Care and Use Committee. Mice were housed in an approved UNC animal care facility on a 12-hour light/dark cycle with ad libitum food and water access. Immediately before the terminal surgery, mice were transported to the research laboratory, where they were deeply anesthetized with sodium pentobarbital (80 mg/kg ip). JRC Immunohistochemistry: Mice were housed on a 12-hour light/dark cycle with ad libitum food and water access. Experimental procedures were conducted

according to the National Institute of Health guidelines for animal research and approved by the Institutional Animal Care and Use Committee at Janelia Farm Research Campus. Approved animal protocol is IACUC 11-71.

## Additional files

### Supplementary files

• Supplementary file 1. Antibody controls. Pearson's correlation coefficients from 4 different control experiments are shown. *The comparison between adjacent sections* tests the consistency of staining, as the distribution of targets is very similar on two adjacent ultrathin sections (70 nm thickness). This correlation is influenced by antibody characteristics, but also the size of targets, with smaller targets displaying larger spatial variability from section to section. This could explain the lower R for NF-H which labels axons, as some axons can be very thin (<100 nm). The comparison with an antibody against an *overlapping antigen* is a test for the specificity of staining. The following comparisons were done: MBP/PLP, GABA/GAD2, PV/GABA, NFHch/NFHr, αTub/ac αTub. The lower coefficients for GABA/GAD2 and PV/GABA reflect the fact that GAD2 and PV are present in only a subset of GABA containing structures. Another test for specificity is the comparison with an antibody against a spatially *exclusive antigen*. Values of R around 0 are expected in this case. MBP and PLP (present in the myelin sheath) were each compared with GABA (inside inhibitory neurons); GABA and PV (inhibitory neurons) were compared with VGluT1 (excitatory neurons); NFH, αTub and ac αTub (all predominantly neuronal) with glutamine synthetase (glial). And finally, all antibodies were compared with DAPI to control for background *nuclear* staining.

### Major datasets

The following previously published dataset was used:

| Author(s) | Year | Dataset title | Dataset URL | Database, license, and accessibility information |
|---|---|---|---|---|
| Bock DD, Lee WC, Kerlin AM, Andermann ML, Hood G, Wetzel AW, Yurgenson S, Soucy ER, Kim HS, Reid RC | 2011 | Network anatomy and in vivo physiology of visual cortical neurons | http://openconnecto.me/catmaid/?pid=4&zp=0&yp=59903&xp=67711&tool=navigator&sid0=4&s0=10 | Publicly available at the Open Connectome Project (Mouse V1; http://openconnecto.me/bock11/) |

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
