## [Decision Letter]

Thank you for submitting your article "A large fraction of neocortical myelin ensheathes axons of local inhibitory neurons" for consideration by *eLife*. Your article has been favorably evaluated by Eve Marder (Senior editor) and three reviewers, one of whom, Inna Slutsky, is a member of our Board of Reviewing Editors.

The reviewers have discussed the reviews with one another and the Reviewing Editor has drafted this decision to help you prepare a revised submission.

Summary:

The reviewers judged that this is an interesting manuscript examining myelination of interneurons in cortical grey matter using two complementary microscopy approaches. The authors show that a significant proportion of cortical myelin ensheathes axons of inhibitory neurons and that the vast majority of myelinated inhibitory axons belong to a specific interneuron subtype, the PV-positive basket cell. While early publications in cats and macaques demonstrated myelinated axons in basket cells, this study generalizes these findings to mice and provides additional quantification of postsynaptic partners, myelin distribution and composition in PV-expressing neurons. The study will be of great interest to readers in multiple disciplines and provides information relevant for future physiological studies of activity-dependent myelination and de-/re-myelination of grey matter in various neurodegenerative disorders.

Essential revisions:

1) The comparison of the myelinated axons in the GABAergic vs non-GABAergic axons and myelinated vs non-myelinated parts of PV-(+) cell axons is incomplete. More careful analysis of the data is required for improving the description of myelin protein composition, thickness and cytoskeletal composition of the myelinated inhibitory axons.

Figure 7 and Figure 8 suggest that the inhibitory myelinated axons are larger in diameter than the excitatory myelinated axons, but they do not test or quantify this directly.

The relationship between the relative MBP levels and the properties of axons/myelin sheaths is unclear. There may be methodological concerns, because MBP is notoriously difficult to visualize and requires extraction of lipids for antibodies to get access. Since there is no calibration of the fluorescent intensity, quite what to make of the reported 30% greater intensity on the PV-(+) axons is unclear and the significance of this difference is greatly amplified in the Discussion.

Analysis of the starting and end points of myelination in the inhibitory axons should be provided. Does it follow axonal initial segment as in the excitatory axons and at which branches myelination disappears?

It would be important to understand the differences between myelinated and unmyelinated parts of PV-(+) axons in more details. For example, comparative literature analysis of several parameters such as number/release properties of PV's inhibitory terminals, mitochondrial content etc. between layer 2/3 and layer 5 would be helpful.

How similar is the morphology of oligodendrocytes between inhibitory and excitatory myelinated axons?

Figure 9 does not provide a comparison of cytoskeletal composition between the myelinated and non-myelinated inhibitory axon segments on the same neurons, or between myelinated and unmyelinated inhibitory axons in general. The reported cytoskeletal differences may therefore be related to the inhibitory vs excitatory distinction rather than inhibitory myelinated vs. excitatory myelinated (i.e. inhibitory and excitatory axons may already have that difference regardless of their myelination).

2) The reviewers don't understand a discrepancy between Figure 1. For example, the equal density of GABA and non-GABA myelinated axons in layer 2/3 are shown in D, while 50% of myelinated axons in this layer are GABAergic, as seen in C. But in L4 and L5 the density shown in D does not look that different from L2/3, but in C it does. Do we miss something?

3) Some reviewers feel that the Introduction over-dramatizes the historic difficulties of studying myelin on identified cells. We encourage you to acknowledge the findings by DeFelipe, Hendry and Jones (1986) who used the Golgi stain in their study of large basket cells in the sensory-motor cortex of monkey and showed that the axon became myelinated where it lost its Golgi-staining. Similarly, other combined light and electron microscopic methods have been available for at least 30 years to analyze the individual myelinated axonal arbors.

4) The Discussion is over-long and over-speculative. The basic result showing that PV-(+) basket cells have myelinated axons is consistent with observations in other species. Too much speculation is devoted to what this might mean. Particularly speculative is the section on the clinical significance of the findings.

---

## [Author Response]

*Essential revisions:*

*1) The comparison of the myelinated axons in the GABAergic vs non-GABAergic axons and myelinated vs non-myelinated parts of PV-(+) cell axons is incomplete. More careful analysis of the data is required for improving the description of myelin protein composition, thickness and cytoskeletal composition of the myelinated inhibitory axons.*

Figure 7 and Figure 8 suggest that the inhibitory myelinated axons are larger in diameter than the excitatory myelinated axons, but they do not test or quantify this directly.

In the Results section, we state, “GABA axons are thicker than non-GABA axons (0.54 ± 0.001 µm vs. 0.45 ± 0.001 µm, n= 163 GABA and 238 non-GABA axons, P<0.0001, Mann-Whitney U Test)” (subsection “GABA and non-GABA myelin differ in protein composition, but not thickness”, first paragraph). To make this point clearer, we have added a panel to Figure 8 (panel A, showing the frequency distribution of axon thickness for GABA and non-GABA axons).

*The relationship between the relative MBP levels and the properties of axons/myelin sheaths is unclear. There may be methodological concerns, because MBP is notoriously difficult to visualize and requires extraction of lipids for antibodies to get access. Since there is no calibration of the fluorescent intensity, quite what to make of the reported 30% greater intensity on the PV-(+) axons is unclear and the significance of this difference is greatly amplified in the Discussion.*

We agree that the 30% greater intensity of MBP immunofluorescence in GABA-positive axons compared to their neighboring GABA-negative axons is uncalibrated. We have accordingly de-emphasized the importance of the quantitative difference in Results (subsection “GABA and non-GABA myelin differ in protein composition, but not thickness”, last paragraph), instead making it clear that the difference is qualitative. We have also shortened the corresponding discussion on the topic, for example, by deleting the speculation that “Our observation that inhibitory myelin has a higher content of MBP, a potential target antigen in MS^92-94^, suggests that inhibitory axons within the RRMS lesions might be more susceptible to demyelination than excitatory axons.”

We would like, however, to point out that the quality of MBP immunostaining with array tomography (AT) is exceptionally good, as evidenced, for example, by the correlation (R^2^=0.9) of MBP immunofluorescence and ultrastructurally identified myelin (Figure 1—figure supplement 2). The ultrathin 70 nm sections used in AT likely provide superior access to the MBP antigen in the myelin; additionally, the freeze-substitution approach to tissue preparation aids in the preservation of the antigen^1,2^. To further illustrate the high quality of the MBP immunofluorescence in AT, we have now added a panel to Figure 8, showing a higher magnification view of MBP immunofluorescence, and a supplemental image stack of raw images of serial sections through mouse cortex immunostained for MBP (Video 1). We have also added text in the 2^nd^ paragraph of the Results. As can be seen in Video 1, MBP immunofluorescence around individual axons is present as a clear and uniform outline, which is consistent from section to section. We have quantified the consistency of MBP immunolabel by comparing the MBP signal from individual myelin sheaths on adjacent sections (added to Figure 1—figure supplement 2, panel B).

*Analysis of the starting and end points of myelination in the inhibitory axons should be provided. Does it follow axonal initial segment as in the excitatory axons and at which branches myelination disappears?*

Unfortunately, our anatomical data do not provide a good sample of the distribution of myelin on the complete axonal arbors of inhibitory neurons, as noted in the ‘caveats and limitations’ section of the Discussion (fifth and sixth paragraphs), and we therefore cannot systematically address this important question in the current study. However, in the cases where a myelinated inhibitory axon was traced back to a cell body in the AT and EM data sets, we noted that the axon “becomes myelinated soon after exiting the cell body (usually within 20-50 μm)” (subsection “Inhibitory neurons exhibit a distinct pattern of myelination”), consistent with myelination commencing shortly after the axon initial segment. We have edited the text at this point to highlight this relationship. Regarding the broader distribution of myelin, we note a qualitative impression, that “All of the axonal arbors that were traced back to the cell body in our volume EM dataset (n=8) had the same general appearance: a central core of partially myelinated axons whose unmyelinated stretches rarely made synapses, from which emerged unmyelinated branches forming numerous synapses (Figure 2, Figure 5). This is similar to the pattern reported for filled basket cells in both macaque somatosensory cortex (DeFelipe, Hendry and Jones, 1986) and cat visual cortex (Somogyi et al., 1983).”

*It would be important to understand the differences between myelinated and unmyelinated parts of PV-(+) axons in more details. For example, comparative literature analysis of several parameters such as number/release properties of PV's inhibitory terminals, mitochondrial content etc. between layer 2/3 and layer 5 would be helpful.*

We could not find published data comparing myelinated and unmyelinated regions of PV+ axons. We agree that the difference between the myelinated and unmyelinated parts of PV+ axons is an important question, which we did not address in our original submission. We therefore performed additional analysis on this topic and added panel (D) to Figure 9. This addition is discussed further below.

*How similar is the morphology of oligodendrocytes between inhibitory and excitatory myelinated axons?*

This very interesting question is related to the larger and still unsettled issue of whether individual oligodendrocytes or possible oligodendrocyte subtypes discriminate between different axons (for example discussed in de Hoz and Simons, 2015). Although array tomography could be used to address this topic in future studies, for example by adding an oligodendrocyte marker to visualize these cells alongside markers for myelin (MBP) and axonal type (GABA, PV), it is in our opinion a significantly different research focus from the one described in this article. However, we have added a sentence to the Discussion raising this question as a possible avenue for future work (eleventh paragraph).

Figure 9 does not provide a comparison of cytoskeletal composition between the myelinated and non-myelinated inhibitory axon segments on the same neurons, or between myelinated and unmyelinated inhibitory axons in general. The reported cytoskeletal differences may therefore be related to the inhibitory vs excitatory distinction rather than inhibitory myelinated vs. excitatory myelinated (i.e. inhibitory and excitatory axons may already have that difference regardless of their myelination).

As mentioned above we performed additional analysis of our AT data to address this question (added Figure 9, panel D; added to [Supplementary-material SD4-data]; and a new paragraph in Results, last paragraph). For this analysis, we identified individual axons that contained both a myelinated and a non-myelinated portion within the dataset volume. Nodes of Ranvier were excluded and only non-myelinated stretches longer than 4 µm were considered. This analysis revealed several interesting differences. First, as the reviewers suggested, there are indeed cytoskeletal differences that are related to the inhibitory vs. excitatory distinctions. Both the myelinated and the unmyelinated regions of the inhibitory axons have higher neurofilament content and lower microtubule content compared to excitatory axons. In addition, however, there are also statistically significant differences in the cytoskeletal content between the different regions of PV+ axons, where myelinated portions have higher neurofilament and lower microtubule content compared to the unmyelinated portions of the same axons. While excitatory axons show similar trends of differences in cytoskeletal composition depending on myelination, the differences are more pronounced for PV+ axons, in particular for neurofilament content.

2) The reviewers don't understand a discrepancy between Figure 1. For example, the equal density of GABA and non-GABA myelinated axons in layer 2/3 are shown in D, while 50% of myelinated axons in this layer are GABAergic, as seen in C. But in L4 and L5 the density shown in D does not look that different from L2/3, but in C it does. Do we miss something?

We forgot to mention that in Figure 1 the y-axis scale is logarithmic. We have now added this to the figure legend, and thank the reviewers for helping catch this omission.

3) Some reviewers feel that the Introduction over-dramatizes the historic difficulties of studying myelin on identified cells. We encourage you to acknowledge the findings by DeFelipe, Hendry and Jones (1986) who used the Golgi stain in their study of large basket cells in the sensory-motor cortex of monkey and showed that the axon became myelinated where it lost its Golgi-staining. Similarly, other combined light and electron microscopic methods have been available for at least 30 years to analyze the individual myelinated axonal arbors.

Although the work of DeFelipe, Hendry and Jones was cited both in the Introduction and Discussion of the original manuscript, we agree that the text of the Introduction overstated the difficulty of studying myelin on individual filled cell axons and understated the impact the earlier studies have had. We have edited the Introduction accordingly (third paragraph) and included more references of earlier studies.

*4) The Discussion is over-long and over-speculative. The basic result showing that PV-(+) basket cells have myelinated axons is consistent with observations in other species. Too much speculation is devoted to what this might mean. Particularly speculative is the section on the clinical significance of the findings.*

We have contracted the Discussion, especially the portions on cell physiology (ninth paragraph) and clinical significance (eleventh paragraph).

References

1) McDonald, K. High-pressure freezing for preservation of high resolution fine structure and antigenicity for immunolabeling. Methods Mol Biol 117, 77-97, (1999).

2) Hippe-Sanwald, S. Impact of freeze substitution on biological electron microscopy. Microsc Res Tech 24, 400-422, (1993).